# Missingness-MDPs: Bridging the Theory of Missing Data and POMDPs

## Abstract

We introduce *missingness-MDPs* (miss-MDPs); a subclass of partially observable Markov decision processes (POMDPs) that incorporates the theory of missing data. Miss-MDPs capture settings where, at each step, features of the current state may go missing, that is, the state is not fully observed. Missingness of state features occurs dynamically, governed by the *missingness function*, a restricted observation function. In Miss-MDPs, we distinguish three types of missingness functions: missing completely at random (MCAR), missing at random (MAR), and missing not at random (MNAR). Our problem is to compute a policy for a miss-MDP with an *unknown* missingness function from a dataset of **observations and actions**. We propose probably approximately correct (PAC) algorithms that, from a dataset, approximate the missingness function and, thereby, the true miss-MDP. We show that, for specific missingness functions, the policy computed on the approximated model is $\varepsilon$-optimal in the true miss-MDP. The empirical evaluation confirms these findings and shows that our approach becomes more sample-efficient when exploiting the type of the missingness function.

## 1 Introduction

Markov decision processes (MDPs; Puterman, 1994) capture sequential decision-making under uncertainty. Classically, it is assumed that all *state features* can be precisely measured at all times. However, such features can be *missing*, e.g. due to sensor failure, so decisions cannot be made based on all features. Consider a medical doctor diagnosing a patient based on the state features of heart rate and temperature: Such measurements might be incomplete.

Partially observable Markov decision processes (POMDPs; Åström, 1965) can capture the aspect of missing state features. In POMDPs, an *observation function* explicitly models uncertainty in the observations of state features, and policies are based on the resulting beliefs over features. A policy thus describes how an agent (or doctor) should act given its current belief. Yet, solving POMDPs is notoriously challenging: In particular, inferring the observation function from observations of features alone is generally *intractable* as the probabilities depend on the past sequences of actions and observations (Liu et al., 2022a; Lee et al., 2023).

Fortunately, specific problems often exhibit a simpler structure in the source of partial observability: The *missingness* of state features may occur according to a stochastic *missingness function*. Such problems are studied by the theory of *missingness* (Schafer & Graham, 2002; Buuren, 2018; Little & Rubin, 2019). As practical reasons for missingness vary, Rubin (1976) classifies missingness functions into three main types: *missing completely at random* (MCAR), *missing at random* (MAR), and *missing not at random* (MNAR). MCAR missingness is independent of observed or unobserved state features – e.g., the temperature feature is missing due to a loosely attached thermometer. MAR missingness solely depends on observed state features – e.g., the observed temperature feature influences the missingness of the heart rate feature. Missingness functions that are neither MCAR nor MAR are considered MNAR – e.g., the temperature feature influences its own missingness.

Prior work on sequential decision-making with missing observations has focused mainly on reinforcement learning, where missing data are treated as incidental rather than explicitly modeled (Lizotte et al., 2008; Li et al., 2018; Wang et al., 2019; Böck et al., 2022). Planning approaches typically overlook distinctions between MCAR, MAR, and MNAR (Liu et al., 2022b; Yamaguchi et al., 2020; Futoma et al., 2020), or rely on implicit assumptions about the missingness mechanism, which can

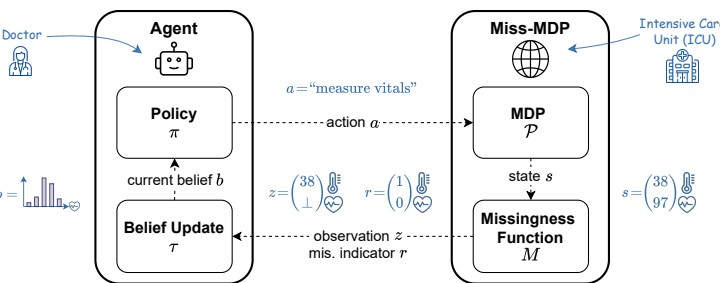

Figure 1: A doctor-treating-patient example (blue annotations) of an agent interacting with a miss-MDP. The missingness function causes the heart rate feature of the state to go missing, indicated as $\perp$ in the observation. The missingness indicator evaluates to 0 for missing features and to 1 otherwise.

lead to biased or inconsistent estimates (Futoma et al., 2020) and provide no guarantees on policy performance (Yamaguchi et al., 2020). To our knowledge, no existing work bridges missingness and POMDPs to (1) explicitly model and learn the missingness function up to statistical guarantees, and (2) leverage the learned function to guarantee the optimality of the resulting policies.

To formalize missing state features in MDPs, we define *missingness-MDPs* (miss-MDPs) as a proper subclass of POMDPs. In miss-MDPs, the observation function is a missingness function. This function, categorized as MCAR, MAR, or MNAR, explicitly induces missing state features in the observations. In Figure 1 we depict a doctor-treating-patient example with a miss-MDP. The problem is as follows: given (1) a miss-MDP with an *unknown* missingness function and (2) a dataset of observations sampled from the miss-MDP, the goal is to compute a belief-based policy that maximizes the expected reward. To obtain guarantees on the result, our approach is to approximate a missingness function from the dataset, and thereby approximate the original miss-MDP. For this approximate miss-MDP, we compute a policy through off-the-shelf POMDP solvers such as SARSOP (Kurniawati et al., 2008). Missingness functions are *not learnable in general* (Bhattacharya et al., 2020), yet we identify and, subsequently, focus on missingness functions that are tractable to learn.

In summary, our contributions are:

1. We introduce miss-MDPs, which integrate and define the semantics of missingness in a specific subclass of the more general POMDP framework (Section 3).

2. We identify that beliefs over state features do not always depend on the probabilities of the missingness function (Remark 1), similar to *ignorability* of missing data (Little & Rubin, 2019).

3. We provide algorithms with *probably approximately correct* (PAC) guarantees for tractable subsets of the three main types of missingness functions (Sections 4.1 and 4.2).

4. Using these algorithms, we prove that we can approximate the $\varepsilon$-optimal policy for the miss-MDP under the correct assumption on the missingness function (Section 4.3).

Our empirical evaluation (Section 5) confirms our theory and highlights the practical advantages of our approach: Using datasets of reasonable size, the performance of policies computed using the learned missingness function converges to that of the optimal policy.

RELATED WORK

Our work builds on a rich literature in missing data analysis, see e.g. (Tsiatis, 2006; Little & Rubin, 2019). Classical assumptions such as MCAR, MAR, and MNAR provide high-level categories. More refined tools, such as missingness graphs, allow one to encode assumptions about the missingness in a structured way (Mohan et al., 2013; Shpitser et al., 2015), leading to highly specific learnability results (Bhattacharya et al., 2020; Nabi et al., 2020). Our setting departs from the standard missing data paradigm in several important aspects. In particular, the concept of missingness is embedded within the broader POMDP setting, which allows for a better and principled understanding of missingness in the context of sequential decision-making under uncertainty.

As noted previously, most work on decision making with missing data focuses on RL, where either full observations (Chen et al., 2023) or individual features may be missing (Shim et al., 2018; Yoon et al.,

2019; Böck et al., 2022). Some approaches incorporate missingness into belief updates for RL agents (Wang et al., 2019), while others adopt model-based methods, often restricted to simpler settings such as MCAR (Futoma et al., 2020). Another line of work combines deep learning with POMDP solvers by learning abstract state representations, but without explicitly modeling the missingness process (Liu et al., 2022b). More principled imputation strategies—such as Bayesian multiple imputation (Lizotte et al., 2008) and expectation-maximization (Yamaguchi et al., 2020)—estimate missing values as an intermediate step in policy computation. In contrast to imputation, our approach directly learns the missingness function and offers PAC guarantees on the resulting policy.

## 2 PRELIMINARIES

A function $\mu\colon X \to [0,1]$ is a *probability distribution over a countable set* $X$ when $\sum_{x\in X}\mu(x) = 1$. The set of such distributions is $\Delta(X)$. The *support* of distribution $\mu \in \Delta(X)$ is $\mathrm{supp}(\mu) = \{x \in X \mid \mu(x) \neq 0\}$. Writing $\mu = \{x_1 \mapsto p_1, \ldots, x_k \mapsto p_k\}$ indicates that $\mu(x_1) = p_1$ and so on. The random variable $x$ sampled from $\mu$ is denoted by $x \sim \mu$. Given $\sigma\colon X \to \Delta(Y)$, we let $\sigma(y \mid x) := \sigma(x)(y)$. The indicator function $\mathbf{1}_\varphi$ returns 1 if predicate $\varphi$ holds and 0 otherwise.

**Definition 1** (POMDPs). A *partially observable Markov decision process* is a tuple $\mathcal{P} = (S, A, T, b_0, \varrho, Z, O, \gamma)$ with finite factored *state space* $S = \times_{i=1,\ldots,n} S_i$ and the set of *feature indices* $I = \{1,\ldots,n\}$), finite *action space* $A$, *transition function* $T\colon S \times A \to \Delta(S)$, *initial state distribution* $b_0 \in \Delta(S)$, *reward function* $\varrho\colon S \times A \to \mathbb{R}$, finite factored *observation space* $Z = \times_{i=1,\ldots m} Z_i$, *observation function* $O\colon S \to \Delta(Z)$, and *discount factor* $\gamma \in [0,1)$.

**Without loss of generality, we consider the observation function to be action-independent, as the state space of a POMDP can be augmented to carry the information of the last performed action (Chatterjee et al., 2016).**

A *trajectory* in a POMDP $\mathcal{P}$ is a sequence of states, observations, and actions. A *history* $h = \left(z^{(0)}, a^{(0)}, z^{(1)}, a^{(1)}, \ldots\right) \in \mathcal{H} \subseteq (Z \times A)^*$ is the observable fragment of a trajectory, i.e., a sequence of observations and actions. A history can be summarized by a *sufficient statistic* known as a *belief* $b \in \mathcal{B} \subseteq \Delta(S)$; a probability distribution over underlying states induced by a history $h \in \mathcal{H}$. The *belief update* $\tau\colon \mathcal{B} \times A \times Z \to \mathcal{B}$ computes a *successor belief* $b'$ via Bayes' rule (Spaan, 2012).

A *policy* $\pi\colon \mathcal{B} \to \Delta(A) \in \Pi$ maps beliefs to probability distributions over actions. The *objective* is to find a policy $\pi \in \Pi$ that maximizes the infinite-horizon expected cumulative discounted reward: $V_{\mathcal{P}}(\pi) = \mathbb{E}^\pi\left[\sum_{t=0}^\infty \gamma^t \varrho(s^{(t)}, a^{(t)})\right]$. As the problem of finding the optimal policy is undecidable (Madani et al., 2003), we focus on computing $\varepsilon$-optimal policies (Hauskrecht, 2000).

## 3 MISSINGNESS IN MDPS

This section introduces missingness-MDPs and the different types of missingness functions.

**Definition 2** (Miss-MDP). A missingness-MDP is a tuple $(S, A, T, b_0, \varrho, Z, M, \gamma)$, where $S$, $A$, $T$, $b_0$, $\varrho$, and $\gamma$ are as in a POMDP, the finite *observation space* is $Z = \times_{i\in I}(S_i \cup \{\bot\})$, with $\bot$ denoting *missing information*, and function $M\colon S \to \Delta(Z)$ is the *missingness function* such that $\forall s \in S, \forall z \in \mathrm{supp}(M(s)), \forall i \in I$ either $z_i = s_i$ or $z_i = \bot$.

Miss-MDPs are a subclass of POMDPs where the state space $S$ and observation space $Z$ share the feature indices $I$, and where $Z \supsetneq S$ because some features can go *missing* in $Z$, being replaced by the symbol $\bot$. This process of "poking holes" is governed by the stochastic missingness function $M$. **While $M$ may take actions into account, we use an action-independent $M$ w.l.o.g (see Section 2).**

**Missingness indicators.** Missingness functions can equivalently be described as a map to vectors of *missingness indicators* (Mohan et al., 2013), i.e. $M\colon S \to \Delta(R)$, where $R = \{0,1\}^n$. A vector $r \in R$ has $r_i = 0$ if feature $i$ is missing ($z_i = \bot$), and otherwise $r_i = 1$. The function $f_R\colon Z \to R$ maps observations to their missingness indicators.

**Example 1.** Let $\mathcal{P}$ be a miss-MDP with $S = \{a,b\}^2$, $Z = \{a,b,\bot\}^2$, and missingness function defined as: $M((s_1, s_2)) = \{(s_1, s_2) \mapsto 0.5, (s_1, \bot) \mapsto 0.5\}$. Then, visiting state $(b, a)$ yields either $(b,a)$ or $(b,\bot)$, each with probability 0.5. We have $f_R((b,a)) = (1,1)$ and $f_R((b,\bot)) = (1,0)$.

We aim to compute a near-optimal policy for a miss-MDP $\mathcal{P}$ with *unknown* missingness function $M$. For this, we use a dataset $\mathcal{D}$ of histories (of length at least $|S|$), which are collected using a *fair* policy (i.e. it has positive probability to visit all reachable states). The resulting policy is *probably approximately correct* (PAC) if, with high probability, its value is close to the true optimum. Formally:

---

**Problem statement.** We are given a miss-MDP $\mathcal{P}$ with an *unknown* missingness function $M$, a dataset $\mathcal{D} = (h_1, \ldots, h_k)$ of $k$ histories $h_i \in \mathcal{H}$ collected from $\mathcal{P}$ under an unknown but fair policy $\pi_b$, and a precision $\varepsilon > 0$ and confidence threshold $\delta > 0$. The goal is to approximate the missingness function $\widehat{M} \approx M$ for all reachable states and use it to compute a policy $\pi^* \in \Pi$ such that with probability at least $1 - \delta$, we have $\sup_\pi (V_\mathcal{P}(\pi)) - V_\mathcal{P}(\pi^*) \leq \varepsilon$.

---

### 3.1 TYPES OF MISSINGNESS FUNCTIONS

We formally introduce the three types of missingness functions (MCAR, MAR, and MNAR) in the context of miss-MDPs using missingness indicators $r \in R$ (see Section 3). The simplest is MCAR, where the probability of a feature going missing does not depend on any *feature values* of the state. The miss-MDP in Example 1 is of type MCAR.

**Definition 3** (**MCAR**). The missingness function $M \colon S \to \Delta(Z)$ of a miss-MDP is MCAR iff $\forall r \in R, \exists p_r \in [0, 1], \forall s \in S, \mathbb{P}(f_R(\boldsymbol{z}) = r \mid \boldsymbol{z} \sim M(s)) = p_r$.

**Admittability and $I_{\mathrm{always}}$.** We introduce a notion of admittability that indicates whether an observation $z$ could originate from a state $s$. We say that $z$ is *admittable* by $s$, denoted $z \preceq s$, if and only if $\forall i \in I$, $z_i = \bot$ or $z_i = s_i$. In Example 1, we have $(b, \bot) \preceq (b, a)$ and $(b, a) \preceq (b, a)$ but $(a, \bot) \not\preceq (b, a)$. Furthermore, $I_{\mathrm{always}} = \{i \in I \mid \forall s' \in S \colon \mathbb{P}(\boldsymbol{z}_i = \bot \mid \boldsymbol{z} \sim M(s')) = 0\} \subseteq I$ is the set of indices of always observed features, and $I_{\mathrm{mis}} = I \setminus I_{\mathrm{always}}$ is its complement.

We distinguish two MAR variants: a restricted one we call *simple MAR* (Mohan & Pearl, 2021), and the general MAR definition (Rubin, 1976). For simple MAR, the missingness probability of a feature is only influenced by the observable features that never go missing, i.e., by $z_i$ for $i \in I_{\mathrm{always}}$. For MAR, a missingness probability is only influenced by the non-missing features of a given *observation*, including features that may go missing. Any MCAR missingness function is also (simple) MAR.

**Definition 4** (**(Simple) MAR**). The missingness function $M \colon S \to \Delta(Z)$ of a miss-MDP is:

- **Simple MAR** iff for all $s, s' \in S$ that agree on always-observed features (i.e. $\forall i \in I_{\mathrm{always}}$, $s_i = s_i'$), the missingness probability is the same for all missingness indicators $r \in R$, formally: $\mathbb{P}(f_R(\boldsymbol{z}) = r \mid \boldsymbol{z} \sim M(s)) = \mathbb{P}(f_R(\boldsymbol{z}') = r \mid \boldsymbol{z}' \sim M(s'))$.
- **MAR** iff for all $s, s' \in S$ and $z \in Z$, if $z \preceq s, s'$, the probability of its missingness indicator $r \coloneqq f_R(z)$ is equal for both states: $\mathbb{P}(f_R(\boldsymbol{z}') = r \mid \boldsymbol{z}' \sim M(s)) = \mathbb{P}(f_R(\boldsymbol{z}'') = r \mid \boldsymbol{z}'' \sim M(s'))$.

**Example 2.** We redefine $M$ in the miss-MDP from Example 1 to be simple MAR: $M((s_1, a)) = \{(s_1, a) \mapsto 1\}$, and $M((s_1, b)) = \{(s_1, b) \mapsto 0.5, (\bot, b) \mapsto 0.5\}$. Here, the missingness probability of feature 1 depends on the *always* observed value of feature 2. As an example of MAR **which is** not simple MAR, consider: $M((s_1, a)) = \{(s_1, a) \mapsto 0.5, (\bot, \bot) \mapsto 0.5\}$, and $M((s_1, b)) = \{(s_1, b) \mapsto 0.25, (\bot, b) \mapsto 0.25, (\bot, \bot) \mapsto 0.5\}$. **Here, feature 2 may go missing as well. The missingness probability of feature 1 depends on the value of feature 2. But only when it is *observed*!**

**Definition 5** (**MNAR**). The missingness function $M$ of a miss-MDP is MNAR iff it is not MAR.

For MNAR, missingness probabilities may depend on the values of missing features. In particular, in *self-censoring* missingness functions, a feature's missingness probability depends on its own value.

**Example 3.** We adapt Example 1 to make $M$ MNAR and self-censoring for feature 2: $M((s_1, a)) = \{(s_1, a) \mapsto 0.5, (s_1, \bot) \mapsto 0.5\}$ and $M((s_1, b)) = \{(s_1, b) \mapsto 0.1, (s_1, \bot) \mapsto 0.9\}$.

### 3.2 MISSINGNESS GRAPHS

*Missingness graphs* (m-graphs) help visualize the dependencies of missingness functions. We adopt and translate the definition of Mohan & Pearl (2021) to our framework of miss-MDPs. An m-graph

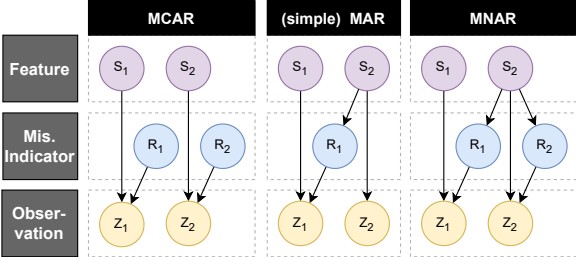

Figure 2: **Example of missingness graphs visualizing relations between the elements of a miss-MDP for the three types of missingness functions.**

is a causal diagram (Pearl, 1995) in the form of a directed acyclic graph. The vertices in the graph correspond to variables, and the directed edges correspond to the relationships between the variables.

The vertices can be grouped into the following categories: $\circledS$-nodes correspond to features of the state space, $\circledZ$-nodes correspond to the features of observations and $\circledR$-nodes correspond to the missingness indicators.[1] For always observed features, we omit the respective $\circledR$-node from the m-graph. Arrows between nodes represent a direct causal relationship: The parent node is a direct cause of the child node. The absence of an edge intuitively denotes that two variables do not directly influence each other; formally, it means that they are conditionally independent, given other variables in the graph according to the d-separation criteria (Pearl, 2009).

**Visualizing types of missingness.** Figure 2 uses m-graphs to illustrate the conditional independence assumptions of different types of missingness functions. For MCAR, both $\circledR$-nodes are purely stochastic, having no incoming arrows and thus not depending on any feature value. For (simple) MAR, there are two changes: Feature $S_2$ is always observable ($R_2$ is absent), and it affects missingness indicator $R_1$ (red arrow). For MNAR, $S_2$ can go missing, so $R_1$ depends on information that can go missing. We remark that m-graphs cannot represent *context-specific* independence assumptions, which are needed to, e.g., represent non-simple MAR functions such as the one in Example 2; but the missingness functions we focus on may all be represented by m-graphs. Further, we provide the corresponding m-graphs for all experiments in Appendix C.

## 4 APPROXIMATING MISSINGNESS-MDPS

Our goal is to compute $\varepsilon$-optimal policies for a miss-MDP. For this, we first compute an approximation $\widehat{M} \approx M$ of the unknown $M$ from the given dataset $\mathcal{D}$ of histories. This yields an approximated, but fully specified miss-MDP $\widehat{\mathcal{P}}$, which can be solved using any off-the-shelf POMDP solution method.

**Missingness types in focus.** A necessary condition is that the missingness function can be approximated solely from observations, a property that missing data literature calls *identifiability* (Bhattacharya et al., 2020). Establishing identifiability is not the focus of this paper. Instead, we provide PAC guarantees for two types that are known to be identifiable. Thus, we focus on: (1) simple MAR (including MCAR), and (2) non-self-censoring MNAR with independent missingness indicators. Additionally, in Section 5, we experiment on MNAR *with* dependencies between the indicators.

**Outline.** Remark 1 presents an interesting insight orthogonal to our problem: For maintaining a belief during policy execution, certain types of missingness can in fact be *ignored*. Sections 4.1 and 4.2 describe our algorithms for approximating missingness functions. Both are structured as follows: They state assumptions, define how to compute $\widehat{M}$, prove that the approximation is probably approximately correct, and explain how to utilize additional knowledge on the missingness function to reduce sample complexity. Section 4.3 uses these algorithms to compute near-optimal policies.

**Remark 1** (Ignorability). Missing data literature defines *ignorability* as cases where any quantity of interest can be consistently estimated from observations alone and it is not necessary to model the

---

[1]We exclude the category of unobserved features $U$ used in Mohan & Pearl (2021), as in our setting $U = \emptyset$ since $M$ depends on states.

missingness process (Little & Rubin, 2019). This holds under MCAR, and also under MAR whenever the quantity depends only on the observed features. We identify a similar notion of ignorability for miss-MDPs: If the missingness function $M$ is MAR (including MCAR), then belief updates $\tau$ can be computed without knowledge of the precise probabilities of $M$, since these cancel out in Bayes' rule; see Appendix A for a formal proof. Thus, MAR missingness is ignorable for maintaining a belief when executing a policy in a miss-MDP. However, we stress that the missingness function *is* required to compute belief-based policies, since the probabilities of successor beliefs depend on it.

**Occurrence counts.** Both algorithms extract the number of occurrences of every observation using the dataset $\mathcal{D} = (h_1, \ldots, h_k)$ of $k$ histories $h_i \in \mathcal{H}$. For each $h_i = \left(z^{(0)}, a^{(0)}, \ldots, z^{(l)}, a^{(l)}\right)$, we denote the $j$-th observation $z^{(j)}$ by $h_i^{(j)}$. The number of occurrences of an observation $z \in Z$ is: $\#_{\mathcal{D}}(z) = \sum_{i=1}^{k} \sum_{j=0}^{|h_i|} \mathbf{1}_{h_i^{(j)} = z}$. For a set $Z' \subseteq Z$, we define $\#_{\mathcal{D}}(Z') = \sum_{z \in Z'} \#_{\mathcal{D}}(z)$.

## 4.1 APPROXIMATING M FOR MCAR AND SIMPLE MAR

If a missingness function is of type simple MAR, we can approximate it using the *approximation for simple MAR* algorithm, AsMAR. The modifications to obtain the algorithm for the more restricted MCAR-type functions, AMCAR, are described at the end of the section.

**Always-observable features.** Based on the dataset $\mathcal{D}$, we partition the feature indices $I$ into those that are always observed and those that can go missing: $\hat{I}_{\text{always}} = \{i \in I \mid \#_{\mathcal{D}}(\{z \in Z \mid z_i = \bot\}) = 0\}$ and $\hat{I}_{\text{mis}} = I \setminus \hat{I}_{\text{always}}$, respectively. Note, this partitioning is based on empirical data ($\hat{I}_{\text{always}} \approx I_{\text{always}}$) and we might misclassify a feature index to be in $\hat{I}_{\text{always}}$ even though it can go missing.

**Computing $\widehat{M}$.** We use the fact that $M$ can be seen as a mapping $S \to \Delta(R)$ (see paragraph "Missingness indicators", Section 3). Consequently, for every state, we want to approximate the probability of a certain vector of missingness indicators. The simple MAR assumption tells us that the probabilities can only depend on the features in $\hat{I}_{\text{always}}$. Thus, for every combination of the always-observable features of a state $s \in S$ and missingness indicator vector $r \in R$, we can compute the occurrence count $\#_{\mathcal{D}}(s, r) = \#_{\mathcal{D}}\left(Z_s^r\right)$, where $Z_s^r$ is defined as:

$$Z_s^r = \left\{ z \in Z \mid \forall i \in I \colon (i \in \hat{I}_{\text{always}} \implies z_i = s_i) \land (r_i = 0 \implies z_i = \bot) \right\}.$$

Using this, we obtain $\widehat{M}(z \mid s)$ as the fraction of observing $(s, f_R(z))$ and the sum of counts for $s$ and all possible missingness indicators values:

$$\widehat{M}(z \mid s) = \frac{\#_{\mathcal{D}}(s, f_R(z))}{\sum_{r \in R} \#_{\mathcal{D}}(s, r)}. \tag{1}$$

**Probably approximately correct.** With enough data, our approach yields an arbitrarily precise approximation of the true missingness function. We formalize this in Theorem 1 as a PAC guarantee, not only proving that it becomes $\varepsilon$-precise for every $\varepsilon > 0$, but that we can also bound the probability of an error (through unlucky sampling). Additionally, we can adapt the claim to bound the imprecision of the resulting $\widehat{M}$ for a given dataset. The proof is provided in Appendix B.2.

**Theorem 1** (PAC guarantee for AsMAR). Let $\mathcal{P}$ be a missingness-MDP where the missingness function is simple MAR. For every given precision $\varepsilon$ and confidence threshold $\delta$, there exists a number $n^*$ of histories, such that a dataset $\mathcal{D}$ of $n^*$ histories has the following property: With probability at least $\delta$, $\widehat{M}$ computed on $\mathcal{D}$ according to Equation (1) satisfies that for all reachable states $s \in S$ and observations $z \in Z$, we have $|\widehat{M}(z \mid s) - M(z \mid s)| \leq \varepsilon$. Dually, given a dataset $\mathcal{D}$ and confidence threshold $\delta$, we can compute an $\varepsilon$ such that with probability at least $\delta$, for all reachable states $s \in S$ and observations $z \in Z$, we have the same inequality, i.e. $|\widehat{M}(z \mid s) - M(z \mid s)| \leq \varepsilon$.

**Using additional assumptions on the missingness function.** Beyond the necessary simple MAR assumption, we can exploit additional assumptions to improve the approximation of $\widehat{M}$ for the same $\mathcal{D}$. Consider a feature $i$ that is always observable, but does not affect the missingness probability of other features. We can exclude such $i$ from $\hat{I}_{\text{always}}$, effectively merging the occurrence counts of states that differ only in this feature. Therefore, if we assume $M$ to be MCAR, $\hat{I}_{\text{always}}$ can be reduced to an empty set. Consequently, we get that $\#_{\mathcal{D}}(s, r)$ does not depend on $s$ anymore, and we

effectively only count occurrences of missingness indicators, resulting in the algorithm `AMCAR`. We prove the correctness of these improvements in Appendix B.2. In Section 5, we empirically show that using such knowledge can significantly improve the precision of $\widehat{M}$ estimated from the same $\mathcal{D}$.

## 4.2 APPROXIMATING M WITH INDEPENDENT MISSINGNESS INDICATORS

This section presents the *approximation for independent missingness indicators* algorithm, `AIMI`. Its assumptions **on $M$ correspond to a subset of identifiable MNAR missingness functions and** are as follows:

1. **Independence of missingness indicators:** The fact that one feature is missing must not influence the missingness-probability of any other feature. Formally, for $s \in S$ and $z \in Z$, $\mathbb{P}(\boldsymbol{z} \mid \boldsymbol{z} \sim M(s)) = \Pi_{i \in I} \mathbb{P}(\boldsymbol{z}_i \mid \boldsymbol{z} \sim M(s))$.

2. **No self-censoring:** Intuitively, a feature may not influence its own missingness probabilities. Formally, for all $i \in I$ and every pair of states $s, s' \in S$ that differ only in the $i$-th feature ($s_i \neq s'_i$, but for all $j \neq i$ we have $s_j = s'_j$) we have $\mathbb{P}(\boldsymbol{z}_i = \bot \mid \boldsymbol{z} \sim M(s)) = \mathbb{P}(\boldsymbol{z}_i = \bot \mid \boldsymbol{z} \sim M(s'))$.

3. **Positivity:** Intuitively, if a feature affects the missingness probabilities of other features, we need to observe its value to learn the missingness probabilities. However, this is impossible if it always misses. Therefore, we require a *positivity assumption* (Hernán & Robins, 2020): For all $i \in I$ and $s \in S$, we have $\mathbb{P}(\boldsymbol{z}_i \neq \bot \mid \boldsymbol{z} \sim M(s)) > 0$.

**Computing $\widehat{M}$.** We compute the occurrence count for every state $s \in S$, feature $i \in I$ and value of a corresponding $i$-th missingness indicator $r_i \in \{0, 1\}$ as $\#_{\mathcal{D}}(s, i, r_i) = \#_{\mathcal{D}}(Z_s^{i, r_i})$, where $Z_s^{i, r_i}$ is the following set of observations:

$$Z_s^{i, r_i} = \{z \in Z \mid \forall j \in I \setminus \{i\} \colon (z_j = s_j) \wedge (r_i = 0 \iff z_i = \bot)\}.$$

By positivity, a large enough dataset almost surely contains observations to make the counters non-zero (i.e. for all $s$ and $i$, we have $\#(s, i, 0) + \#(s, i, 1) > 0$). The probability of a non self-censoring feature $i$ depends only on the other features $j \in I \setminus \{i\}$. Finally, using the independence assumption, we can infer $\widehat{M}$ by taking the product of the individual missingness probabilities of all features (again viewing $M$ as a mapping $S \to \Delta(R)$, see Section 3):

$$\widehat{M}(z \mid s) = \prod_{i \in I} \frac{\#_{\mathcal{D}}(s, i, f_R(z)_i)}{\#_{\mathcal{D}}(s, i, 0) + \#_{\mathcal{D}}(s, i, 1)}. \tag{2}$$

**Probably approximately correct.** In Appendix B.3, we prove Theorem 2 that provides the same kind of guarantee as in Theorem 1; the only difference are the assumptions on the missingness function and the approach for calculating $\widehat{M}$.

**Theorem 2** (PAC guarantee for `AIMI`). *Let $\mathcal{P}$ be a missingness-MDP where the missingness function satisfies independence, non-self-censoring, and positivity. Then, $\widehat{M}$ computed using Equation (2) offers the same PAC guarantees as specified in Theorem 1.*

**Using additional assumptions on the missingness function.** In its general form, `AIMI` maintains a counter for every combination of the feature valuations of other features $j \in I \setminus \{i\}$. If we know that a certain feature $j$ does not affect the missingness probability of $i$ – there is no edge between the $j$-th Ⓢ-node and the $i$-th Ⓡ-node – we merge the counters for all values of the $j$-th feature. This knowledge comes from **(a)** an m-graph, **(b)** assuming simple MAR while observing feature $j$ goes missing in $\mathcal{D}$, or **(c)** assuming MCAR, in which case we drop the dependency on $s$ in the counters. We prove in Appendix B.3 that all these modifications retain the PAC guarantees.

## 4.3 COMPUTING A POLICY WITH THE APPROXIMATIONS

We show in Appendix B.4 that after finitely many samples, $\widehat{M}$ is accurate enough to yield an $\varepsilon$-optimal policy. We highlight that learning $\widehat{M}$ to precision $\varepsilon$ is insufficient, as the errors in $\widehat{M}$ aggregate when solving the miss-MDP.

**Theorem 3** (Computing $\varepsilon$-optimal Policies). *Let $\mathcal{P}$ be a miss-MDP with a missingness function that is simple MAR or that satisfies independence, no self-censoring, and positivity. Assume we can*

sample histories collected under a fair policy, and we know a lower bound on the smallest missingness probability $p \leq \min_{s \in S, z \in Z} M(z \mid s)$. Then, for every given precision $\varepsilon$ and confidence threshold $\delta$, we can in finite time compute a policy $\pi^*$ such that with probability at least $\delta$ it is $\varepsilon$-optimal, i.e. $(\sup_\pi V_\mathcal{P}(\pi)) - V_\mathcal{P}(\pi^*) \leq \varepsilon$.

Note that we use the notion of PAC guarantee that is common in statistical model checking (Brázdil et al., 2025; Ashok et al., 2019). This is inspired by, but slightly different from the original definition of Valiant (1984), as we return *in finite time* a *policy that performs close to optimal* with high probability.

**Practical considerations.** The guarantees of Theorem 3 concern asymptotic convergence to an $\varepsilon$-optimal policy. Thus, they provide the theoretical foundation of our approach. Still, in practice, the required number of samples is very large, and we work with datasets that are not necessarily sufficient to provide the $\varepsilon$-optimality guarantees. Still, we can infer $\widehat{M}$ from any given dataset and then solve the approximated miss-MDP using an off-the-shelf POMDP solver. For datasets of limited size, we encounter a practical problem: For an observation $z$ with $\#_\mathcal{D}(s, f_R(z)) = 0$, for any $s \in S$ we obtain $\widehat{M}(z \mid s) = 0$, leading to a division by zero for $s$ when performing the belief update $\tau$. We circumvent this case by setting $\#_{\mathcal{D},\kappa}(s, r) = \#_\mathcal{D}(s, r) + \kappa$, i.e. we add a small $\kappa > 0$ to every count. The influence of $\kappa$ diminishes with an increasing dataset size $|\mathcal{D}|$.

## 5 EXPERIMENTS

Our empirical study addresses the following questions:

**Q1.** Do the proposed methods provide adequate approximations of the missingness function?

**Q2.** How do (in)correct assumptions on the missingness function affect the approximation?

**Q3.** As the amount of data increases, does the value of the policy computed on the approximated miss-MDP converge to the optimal value of the true miss-MDP?

**Q4.** How does the value computed from the approximated miss-MDP compare against baselines that do not estimate the missingness function?

**Benchmarks.** We consider two environments with varying types of missingness: (1) *ICU*, a benchmark that models a doctor treating a patient, whose vital measurements are not always available (Johnson et al., 2022), and (2) *Predator*, a variant of the Tag benchmark (Pineau et al., 2003), where a predator is chasing a partially hidden prey. To answer Q2, we consider for our benchmarks a selection of the following four missingness functions: (1) *MCAR*, (2) *sMAR*, a simple MAR function, (3) *MNAR (id.)*, an identifiable MNAR function without self-censoring that satisfies the positivity assumption, and (4) *MNAR (unid.)*, an unidentifiable MNAR function with self-censoring. In the *Predator* benchmark, for all missingness functions, the $(x, y)$-coordinates of the prey can only go missing jointly, i.e. the missingness indicators are dependent; in the *ICU* benchmark, the missingness indicators are always independent. For details on the benchmarks, see Appendix C.

**Protocol, algorithms, and baselines.** For a range of dataset sizes $|\mathcal{D}|$, we collect data using the uniform random policy $\pi^{\text{rnd}}$ where $\forall a \in A, \pi^{\text{rnd}}(a \mid \cdot) = 1/|A|$, and compute the estimate $\widehat{M} \approx M$ using our proposed algorithms: AMCAR (●), AsMAR (▲), and AIMI (■) (Section 4). Each $\widehat{M}$ yields an approximated miss-MDP $\widehat{\mathcal{P}}$, for which we compute a policy $\hat{\pi}$ using the POMDP solver SARSOP (Kurniawati et al., 2008). To assess the efficacy of our approach, we consider the following baselines: **(1)** *optimal*: the SARSOP policy $\pi^*$ computed for the true $M$ (the upper bound); **(2)** *uniform M*: the SARSOP policy $\pi^{M_u}$ computed for $M_u$, a guess of $M$ that is uniform, where every feature independently goes missing with probability 0.5.

**Metrics.** For every dataset size and method, we perform 20 independent runs and report the average together with the interquartile range (shaded area) of the following metrics:

1. To assess the quality of the approximation $\widehat{M}$ compared to $M$ for a miss-MDP $\mathcal{P}$, we compute the *total variation* (TV) of the distributions at a state $s \in S$ as $TV(s) = \frac{1}{2} \sum_{z \preceq s} \left| \widehat{M}(z \mid s) - M(z \mid s) \right|$. We aggregate the TV across states by the average TV (ATV): $1/|S| \sum_s TV(s)$, and the worst TV (WTV): $\max_s TV(s)$.

2. We asses how the various $\hat{\pi}$ from the algorithms perform on the true miss-MDP $\mathcal{P}$ by comparing their value $V_{\mathcal{P}}(\hat{\pi})$ to $V_{\mathcal{P}}(\pi^{M_u})$ and the optimum $V_{\mathcal{P}}(\pi^*)$. All policy values are normalized s.t. 1 and 0 correspond to the values of the *optimum* and *uniform* baselines, respectively.

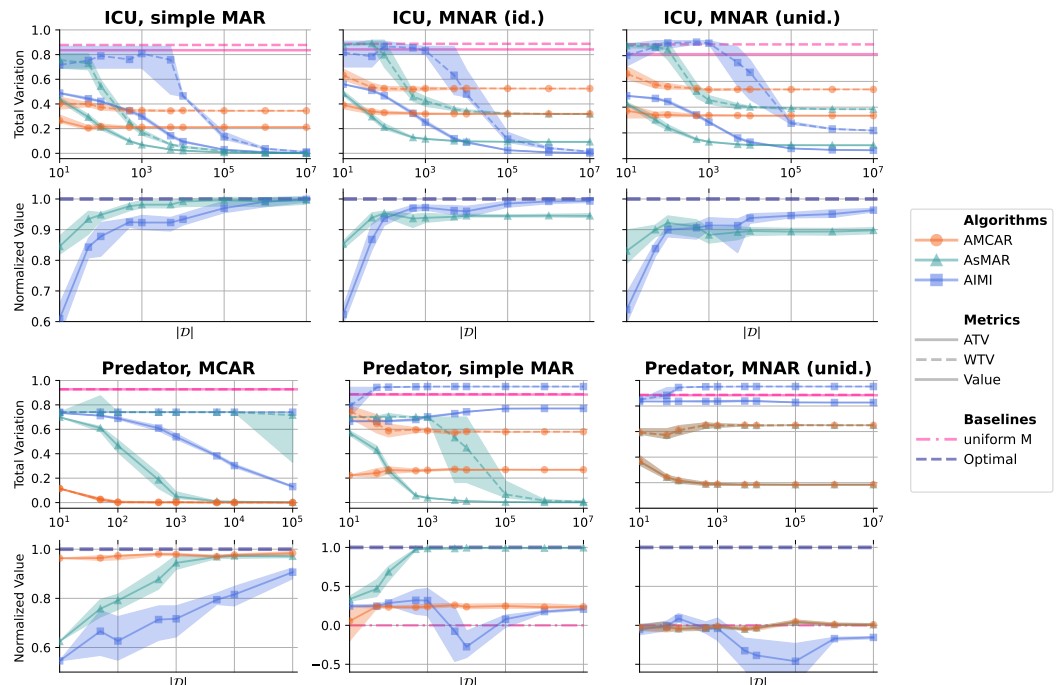

Figure 3: Empirical results for the *ICU* (top) and *Predator* (bottom) benchmarks, including average/-worst total variation (ATV/WTV) and normalized policy values. Values are normalized such that 1 and 0 correspond to the optimal policy (using true $M$) and the uniform baseline, respectively.

**Results.** Figure 3 presents the experimental evaluation. It shows how the TV of $\widehat{M}$ and the value of the associated $\hat{\pi}$ evolve with dataset size $|\mathcal{D}|$. Next, we discuss the questions based on these results.

**Q1: With a sufficient amount of data and the correct assumptions, the algorithms adequately approximate the missingness function.** We observe that under the appropriate assumptions, each algorithm can learn the corresponding missingness function (bringing the TV to zero): AMCAR learns the exact missingness function in *Predator$_{MCAR}$* within 100 observations. We observe similar results for AsMAR (in *ICU$_{sMAR}$* and *Predator$_{sMAR}$*), as well as for AIMI (in *ICU$_{MNAR (id.)}$*).

**Q2: The assumptions on the missingness function significantly affect the quality of the approximation.** On the one hand, relaxing the assumptions on the missingness function ensures it can be learned, though this comes at the cost of reduced sample efficiency. For example, in *Predator$_{MCAR}$*, we observe that AsMAR and AIMI require orders of magnitude more data to learn the missingness function than AMCAR. On the other hand, making stronger assumptions can lead to failures: for example, AMCAR converges to an incorrect missingness function in all benchmarks except *Predator$_{MCAR}$*. The results also show that in some cases, the algorithms might approximate the missingness function even if it does not satisfy the assumptions required for PAC guarantees, as demonstrated from the results of AIMI on *ICU$_{MNAR (unid.)}$*.

**Q3: The convergence to the optimal policy follows the quality of the approximation, and, therefore, the convergence of the resulting policy to the optimum.** With a sufficiently accurate approximation, the value of the policy found by using our methods converges to the optimal value.

**Q4: The values of the policies computed by the baseline are not competitive with the values resulting from our methods.** In all cases, the baseline algorithm fails to approximate the true $M$. The produced polices $\pi^{M_u}$ are significantly worse than the ones resulting from our algorithms under correct type assumptions. The baseline is only competitive on *Predator$_{MNAR (unid.)}$*, where our algorithms also fail due to the fundamental challenge of having an unidentifiable missingness process.

## 6 CONCLUSION

We introduce miss-MDPs to integrate the theory of missing data into decision-making under uncertainty. Given a dataset of observations and actions generated from a miss-MDP, we approximate the unknown missingness function, which – under certain assumptions about the missingness function – enables the computation of an $\varepsilon$-optimal policy. We demonstrate that incorrect assumptions about the missingness mechanism can result in misspecified models and suboptimal policies. Interestingly, we show that for certain missingness functions, belief updates can be computed without knowledge of the missingness function, mirroring the notion of ignorability from the missing data literature. Our experiments support the theoretical results and demonstrate the practical benefits of our contributions. Future work will explore lifting the assumption of a known transition function and extending miss-MDPs to the more general setting of miss-POMDPs.

**Reproducibility Statement.** To ensure the reproducibility of our theoretical results, we provide proofs for all formal claims in the appendix, always referring to the corresponding subsection of the appendix after every claim. For the reproducibility of our practical results, we detail the experimental setup – including experiment parameters as well as hardware specifications – in Section 5 and Appendix C. Further, the repository at `https://anonymous.4open.science/r/missingness-pomdps` provides our implementations of our algorithms and the baselines as used in the experiments, all benchmarks, scripts to rerun the experiments, as well as a README file explaining the technical setup, installation process and creation of datasets.

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

## A    PROOFS FOR SECTION 3: IGNORABILITY

**Lemma 1.** If a missingness function $M$ is MAR, then

$$\forall z \in Z, \exists p \in [0,1], \forall s \in S, M(z \mid s) = \mathbf{1}_{z \preceq s} \cdot p.$$

*Proof.* Suppose that $M$ is MAR. The lemma states that $\forall z \in Z, \exists p \in [0,1], \forall s \in S, M(z \mid s) = p$ if $z \preceq s$ and otherwise $M(z \mid s) = 0$. Since $z \not\preceq s$ implies that $M(z \mid s) = 0$, we only need to show that $\forall z \in Z, \exists p \in [0,1], \forall s \in S, z \preceq s \Rightarrow M(z|s) = p$, which directly follows from the MAR assumption. $\qquad\square$

**Remark 2.** Lemma 1 implies that the missingness function can be omitted in the belief update. Let $b \in \mathcal{B}$ be a belief, and let $s' \in S$. Then, for any $a \in A$ and $z \in Z$, it holds that

$$
\begin{aligned}
b'(s') &= \tau(b,a,z)(s') \\
&:= \frac{M(z \mid s') \sum_{s \in S} T(s' \mid s, a) b(s)}{\sum_{s'' \in S} M(z \mid s'') \sum_{s \in S} T(s'' \mid s, a) b(s)} && \text{(By definition of belief update)} \\
&= \frac{\mathbf{1}_{z \preceq s'} \cdot p \sum_{s \in S} T(s' \mid s, a) b(s)}{\sum_{s'' \in S} \mathbf{1}_{z \preceq s''} \cdot p \sum_{s \in S} T(s'' \mid s, a) b(s)} && \text{(By Lemma 1)} \\
&= \frac{\mathbf{1}_{z \preceq s'} \sum_{s \in S} T(s' \mid s, a) b(s)}{\sum_{s'' \in S} \mathbf{1}_{z \preceq s''} \sum_{s \in S} T(s'' \mid s, a) b(s)}. && \text{($p$ cancels out)}
\end{aligned}
$$

Therefore, the probabilities of $M$ do not affect the resulting probabilities of the belief update. In particular, this means that maintaining a belief while executing a miss-MDP does not require knowledge of $M$.

Still, we stress again that one needs $M$ to compute an optimal policy because this requires constructing and solving the belief MDP (see (Russell & Norvig, 2022, Chapter 16.4.1)), which in turn requires knowing the probability $\mathbb{P}(b' \mid b, a)$ of going to a successor belief $b'$ from a current belief $b \in \mathcal{B}$ upon playing action $a \in A$. Concretely, the probability of a successor belief $b' = \tau(b,a,z)$ depends on the probability of $z \in Z$ given $b$ and $a$, which in turn depends on $M$,

$$
\begin{aligned}
\mathbb{P}(b' \mid b, a) &= \sum_{z \in Z} \mathbb{P}(z \mid b, a) \mathbf{1}_{b' = \tau(b,a,z)}, \\
\mathbb{P}(z \mid b, a) &= \sum_{s \in S} b(s) \sum_{s' \in S} T(s' \mid s, a) M(z \mid s').
\end{aligned}
$$

Here, no normalization occurs, and the probabilities of $M$ do not cancel out.

## B    PROOFS FOR SECTION 4: PROBABLY APPROXIMATELY CORRECT

This appendix is about proving that given enough data, we can approximate the missingness function to arbitrary precision $\varepsilon$, or the other way round: we can prove a certain precision $\varepsilon$ for any given dataset $\mathcal{D}$. In both directions, we provide a probabilistic guarantee, i.e. that the result is correct with probability at least $\delta$. The reason the guarantee has to be probabilistic is that our knowledge relies on a sampled dataset, and, intuitively, there always is a chance that we were "unlucky" and received a very unlikely sequence of samples from which we infer a wrong approximation.

**Outline.** First, in Appendix B.1 we recall standard notions from statistics literature: Bernoulli processes and the fact that building on Okamoto's inequality, we can obtain a size for our dataset $\mathcal{D}$ given precision $\varepsilon$ and confidence $\delta$ (or, analogously, obtain a precision $\varepsilon$ given $\mathcal{D}$ and $\delta$). Afterwards, Appendix B.2 and Appendix B.3 provide the proofs of Theorems 1 and 2, respectively, i.e. the guarantees for our algorithms. Moreover, they prove the guarantees for the modified algorithms when using more information about the missingness function. Finally, Appendix B.4 proves Theorem 3, our main result that $\varepsilon$-policies can be computed.

## B.1 Bernoulli processes

**Definition 6** (Bernoulli process Bernoulli (1713), (Dekking et al., 2005, Chapter 4.3))**.** A Bernoulli process is a sequence of binary random variables that are independent and identically distributed. All random variables have probability $p$ to yield a 1, and probability $1 - p$ to yield a 0.

Throughout this appendix, we write $n$ for the length of the sequence of a Bernoulli process, and $k$ for the number of successes, i.e. the number of times it yielded a 1. Moreover, we denote by $\hat{p} = \frac{k}{n}$ the empirical success probability. Okamoto's seminal work proves the following property of estimating $p$ through observing a Bernoulli process:

**Theorem 4** (Okamoto's inequality (Okamoto, 1959, Theorem 1))**.** For a Bernoulli process with $n$ repetitions and $k$ successes and a given precision $\varepsilon$, we have

$$\Pr(\hat{p} - p \geq \varepsilon) \leq \mathrm{e}^{-2 \cdot n \cdot \varepsilon^2} \text{ and } \Pr(p - \hat{p} \geq \varepsilon) \leq \mathrm{e}^{-2 \cdot n \cdot \varepsilon^2}.$$

Combining these, we get that $\Pr(|\hat{p} - p| \geq \varepsilon) \leq 2 \cdot \mathrm{e}^{-2 \cdot n \cdot \varepsilon^2}$, in words: The probability of the estimate $\hat{p}$ being more than $\varepsilon$ away from the true probability $p$ is less than $2 \cdot \mathrm{e}^{-2 \cdot n \cdot \varepsilon^2}$. For our guarantees, we want to be $\varepsilon$-precise with probability at least $\delta$, so the probability of error should be upper bounded by $1 - \delta$.[2] Thus, we require $2 \cdot \mathrm{e}^{-2 \cdot n \cdot \varepsilon^2} \leq 1 - \delta$. Then, we can solve the inequality for $\varepsilon$ or $n$:

$$2 \cdot \mathrm{e}^{-2 \cdot n \cdot \varepsilon^2} \leq 1 - \delta \Leftrightarrow \varepsilon \geq \sqrt{\frac{\ln(\frac{2}{1-\delta})}{2 \cdot n}} \Leftrightarrow n \geq \frac{\ln(\frac{2}{1-\delta})}{2 \cdot \varepsilon^2}. \tag{3}$$

In other words, given two of precision $\varepsilon$, confidence $\delta$, and number of repetitions $n$, we can infer the third. We remark that there exist other inequalities similar to Okamoto's that yield the same result, but with tighter bounds; we refer to (Budde et al., 2025, Section 3) for a discussion. However, as our goal is only to prove the existence of a bound, we choose the conservative Okamoto bound for its easier accessibility.

## B.2 PAC guarantees for AsMAR

**Theorem 1** (PAC guarantee for AsMAR)**.** Let $\mathcal{P}$ be a missingness-MDP where the missingness function is simple MAR. For every given precision $\varepsilon$ and confidence threshold $\delta$, there exists a number $n^*$ of histories, such that a dataset $\mathcal{D}$ of $n^*$ histories has the following property: With probability at least $\delta$, $\widehat{M}$ computed on $\mathcal{D}$ according to Equation (1) satisfies that for all reachable states $s \in S$ and observations $z \in Z$, we have $|\widehat{M}(z \mid s) - M(z \mid s)| \leq \varepsilon$. Dually, given a dataset $\mathcal{D}$ and confidence threshold $\delta$, we can compute an $\varepsilon$ such that with probability at least $\delta$, for all reachable states $s \in S$ and observations $z \in Z$, we have the same inequality, i.e. $|\widehat{M}(z \mid s) - M(z \mid s)| \leq \varepsilon$.

*Proof.* **Proof outline.** We first show that the computation of every $\widehat{M}(z \mid s)$ is related to a Bernoulli process. Then, using the results of Appendix B.1, we can prove the claims of the theorem for individual state-observation pairs. Next, we lift this to all state-observation pairs by distributing the confidence $\delta$. Finally, we individually explain how this yields the two claims of the theorem.

**The Bernoulli process related to $\widehat{M}(z \mid s)$.** Fix a state $s \in S$ and an observation $z \in Z$. Consider the following random variable: Sample a state $s' \in S$ and the corresponding observation $z' \in Z$. Set the random variable to 1 if $\forall i \in I \colon (i \in I_{\text{always}} \implies z'_i = s_i) \wedge (f_R(z)_i = 0 \implies z'_i = \bot)$; set the random variable to 0 if $\forall i \in I \colon (i \in I_{\text{always}} \implies z'_i = s_i)$; and ignore the sampled $(s', z')$ otherwise, i.e. if $\exists i \in I \colon (i \in I_{\text{always}} \wedge z'_i \neq s_i)$. Note that the random variable is 1 exactly when the sample would be counted by $\#_{\mathcal{D}}(s, f_R(z))$, and the sample is not ignored exactly when it would be counted by $\sum_{r \in R} \#_{\mathcal{D}}(s, r)$.

We require that the probability of the random variable being 1 is equal among all sampled state-observation pairs $(s', z')$ that are not ignored by it, and moreover we require this probability to be equal to $M(z \mid s) = M(f_R(z) \mid s) =: p$. To prove this, we use the assumption that $M$ is a simple MAR

---

[2]Note that in this paper, we use $\delta$ as the probability of the estimate being correct, unlike e.g. Budde et al. (2025), where $\delta$ is the probability of an error.

missingness function; thus, we know that for all $s'$ that agree with $s$ on all always observable features (formally: $\forall i \in I \colon (i \in I_{\text{always}} \implies z'_i = s_i))$, we have $p = M(f_R(z) \mid s) = M(f_R(z) \mid s')$.

We have just shown that the random variable we constructed is a Bernoulli process with success probability $p = M(z \mid s)$, with the number of repetitions $n = \sum_{r \in R} \#_{\mathcal{D}}(s, r)$ and the number of successes $k = \#_{\mathcal{D}}(s, f_R(z'))$. Note that the definition of $\widehat{M}$ in Equation (1) is exactly the empirical success probability $\hat{p} = \frac{k}{n}$.

Observe that we do not need a separate Bernoulli process for every state-observation pair: The number of repetitions $\sum_{r \in R} \#_{\mathcal{D}}(s, r)$ is independent of the observation $z$, since that only affects whether it is counted as success or not. Further, it suffices to have one random variable per combination of valuation for the features in $I_{\text{always}}$, since all states that agree on the always observable features yield the same Bernoulli process. Moreover, we do not need to consider every observation $z$ (as this includes observations that do not admit $s$), but rather only every missingness indicator vector $r \in R$. In the following, we still write "Every state-observation pair" instead of "Every pair of set of states that agree on the always observable features and missingness indicator vector", as it is also true and more concise.

**Single state-observation pair.** Consider the Bernoulli process just described for a fixed state-observation pair $(s, z)$. We explain how to use the results of Appendix B.1 towards proving the first and second claim of the theorem:

- First claim: By the third variant of Equation (3), we have that given a precision $\varepsilon$ and confidence threshold $\delta_{s,z}$, we can compute a necessary number of samples $n_{s,z}$ such that we obtain the PAC guarantee for this state-observation pair.

- Second claim: Observe that a given dataset $\mathcal{D}$ corresponds to a number of repetitions of every Bernoulli process. Let $n_{s,z}$ be the number of repetitions for the pair $(s, z)$. Thus, using the second variant of Equation (3), we have that given $\mathcal{D}$ (and thus $n_{s,z}$) and a confidence threshold $\delta_{s,z}$, we can compute a precision $\varepsilon_{s,z}$ such that we obtain the PAC guarantee for this state-observation pair.

**All state-observation pairs.** We can split the given confidence threshold $\delta$ uniformly over all state-observation pairs, i.e. for every $s \in S$, $z \in Z$, we have $\delta_{s,z} = \frac{\delta}{|S| \cdot |Z|}$. Then, by the union bound, the probability of all state-observation pairs being correctly estimated is the sum of all $\delta_{s,z}$, which (since we distributed it uniformly) is $\delta$. By splitting the confidence threshold in this way, we can obtain the PAC guarantee for all state-observation pairs.

**Second claim.** We first provide the full argument for the second claim, as it is simpler. Given the dataset $\mathcal{D}$ and confidence threshold $\delta$, we obtain an $\varepsilon_{s,z}$ for all state-observation pairs. The probability that all of these are correct is at least $\delta$. We obtain the claim by taking the maximum over these, i.e. setting $\varepsilon := \max_{s \in S, z \in Z} \varepsilon_{s,z}$. Then we have that with probability at least $\delta$, for all states $s \in S$ and observations $z \in Z$, we have $|\widehat{M}(z \mid s) - M(z \mid s)| \leq \varepsilon$.

**First claim.** We proceed in two steps: We explain the analogous argument to the second claim, based on an assumption on the dataset. Afterwards, we explain how this assumption on the dataset can be satisfied.

Assume that for every state-observation pair $(s, z)$, the dataset $\mathcal{D}$ contains at least $n_{s,z}$ samples, i.e. the number computed using Equation (3) inserting $\varepsilon$ and $\delta_{s,z}$. Then, analogously to the proof of the second claim, computing $\widehat{M}$ using this dataset satisfies that with probability at least $\delta$, for all states $s \in S$ and observations $z \in Z$, we have $|\widehat{M}(z \mid s) - M(z \mid s)| \leq \varepsilon$.

It remains to show that there exists a number $n^*$ such that a sampled dataset of $n^*$ histories has the required property. For this, we have to spend some of our confidence threshold $\delta$, since we can only guarantee the property with a certain probability; there is the chance that even upon sampling $n^*$ histories, we are unlucky and some state-observation pair has not been sampled often enough. Thus, we split $\delta$ as follows: $\delta_{\mathcal{D}}$ is used to guarantee the property of the dataset, and $\delta_{\widehat{M}}$ is used to guarantee the consequential property of $\widehat{M}$. Thus, $\delta_{s,z}$ above are obtained by uniformly distributing $\delta_{\widehat{M}}$, not

all of $\delta$. Then, by the union bound, the probability that $\mathcal{D}$ has the desired property and that the PAC guarantee holds is $\delta_{\mathcal{D}} + \delta_{\widehat{M}} = \delta$.

We now need to show that there exists an $n^*$ such that a dataset of this size contains the required number of samples with probability at least $\delta_{\mathcal{D}}$. Recall that the dataset is sampled using a fair policy, which means that every state has a positive probability to be visited; thus (assuming that the length of every history is at least as large the number of states in the miss-MDP), there exists a minimum probability $m$ such that every state is visited with at least probability $m$ in every history. Moreover, observe that for a state-observation pair $(s, z)$, the number of samples for its Bernoulli process is at least the number of times $s$ has been visited; this is because a sample is used when it agrees with $s$ on the always observable features. Thus, for every sampled history, we have a probability of at least $m$ to obtain at least one sample for $(s, z)$. This lower bound on the number of samples for $(s, z)$ is binomially distributed with success probability $m$ (Dekking et al., 2005, Chapter 4.3). Thus, there exists a number of histories $n^*$ such that the probability of having at least $n_{s,z}$ samples for $(s, z)$ when sampling at least $n_h$ histories is greater than $\delta_{\mathcal{D}}$. As before, this argument was for a single state-observation pair; thus, $\delta_{\mathcal{D}}$ is also uniformly distributed over all state-observation pairs.

Summarizing the above: There exists a number $n^*$, such that with probability $\delta_{\mathcal{D}}$, a dataset consisting of $n^*$ histories contains at least $n_{s,z}$ samples for every state-observation pair $(s, z)$, where $n_{s,z}$ is the number computed using Equation (3) inserting $\varepsilon$ and $\delta_{s,z}$. Consequently, $\widehat{M}$ using this dataset satisfies that with probability at least $\delta_{\widehat{M}}$, for all states $s \in S$ and observations $z \in Z$, we have $\widehat{M}(z \mid s) = M(z \mid s) \pm \varepsilon$. Together, we can guarantee that probably (with probability at least $\delta = \delta_{\mathcal{D}} + \delta_{\widehat{M}}$), $\widehat{M}$ is approximately correct.

$\square$

**Proposition 1.** The improvements described in Section 4.1 for using knowledge retain the PAC guarantees stated in Theorem 1.

*Proof.* The improvements use the fact that the underlying Bernoulli process in fact does not depend on all features in $I_{\text{always}}$. While it is correct to still split on these variables, obtaining two processes with the same true success probability, we can also merge them.

More formally, observe that if feature $i$ does not affect the missingness probability of other features, for all valuations of feature $i$, the corresponding Bernoulli processes have the same success probability. MCAR missingness functions are the most extreme case of this, where the given state is completely irrelevant and it suffices to have one Bernoulli process per missingness indicator vector. As a side note: Observe that it is indeed necessary to consider every missingness indicator vector and not individual features, since the missingness probabilities need not be independent. $\square$

B.3   PAC GUARANTEES FOR AIMI (SECTION 4.2)

**Theorem 2** (PAC guarantee for AIMI). Let $\mathcal{P}$ be a missingness-MDP where the missingness function satisfies independence, non-self-censoring, and positivity. Then, $\widehat{M}$ computed using Equation (2) offers the same PAC guarantees as specified in Theorem 1.

*Proof.* This proof is analogous to that of Theorem 1: every missingness probability computed by Equation (2) corresponds to the empirical success probability of a Bernoulli process, which allows to apply the results from Appendix B.1. This proof differs in the argument why all states grouped together in the same Bernoulli process have the same success probability, and in the argument why it feasible to sample a dataset of the necessary size.

By the independence assumption, we know that it suffices to learn every individual $\mathbb{P}(\boldsymbol{z}_i \mid \boldsymbol{z} \sim M(s))$ for each $i \in I$. By non self-censoring, we know that this probability depends only on features in $I \setminus \{i\}$. Thus, the counter $\#(s, i, 0)$ counts exactly the successes of a Bernoulli process with success probability $\mathbb{P}(\boldsymbol{z}_i \mid \boldsymbol{z} \sim M(s))$, and $\#(s, i, 1)$ counts the failures.

It only remains to argue that a sufficient dataset can be feasibly obtained. For this, we use the assumption that no feature is missing surely. In other words, every feature has a positive probability to be observed. Thus, every reachable states has a positive probability $m$ to be fully observed. Using this, we can repeat the argument from the proof of Theorem 1. $\square$

**Proposition 2.** The improvements described in Section 4.1 for using knowledge retain the PAC guarantees stated in Theorem 2.

*Proof.* (a) If we know from an m-graph that a particular feature $i$ is not influenced by feature $j$, for all valuations of $j$ the Bernoulli process has the same success probability. Thus, we can merge these Bernoulli processes and ignore feature $j$.

(b) If we know the missingness function is simple MAR and feature $j$ goes missing, we know that it cannot influence the missingness probability of any other feature by definition (Mohan & Pearl, 2021). Then, the proof is the same as in Case (a).

(c) If the missingness function is MCAR, we know that no feature influences the missingness probability of any other feature. Thus, we can repeatedly apply the argument of Case (a) to merge all Bernoulli processes until we have one for every feature. □

### B.4  COMPUTING $\varepsilon$-OPTIMAL POLICIES (SECTION 4.3)

**Theorem 3** (Computing $\varepsilon$-optimal Policies). Let $\mathcal{P}$ be a miss-MDP with a missingness function that is simple MAR or that satisfies independence, no self-censoring, and positivity. Assume we can sample histories collected under a fair policy, and we know a lower bound on the smallest missingness probability $p \leq \min_{s \in S, z \in Z} M(z \mid s)$. Then, for every given precision $\varepsilon$ and confidence threshold $\delta$, we can in finite time compute a policy $\pi^*$ such that with probability at least $\delta$ it is $\varepsilon$-optimal, i.e. $(\sup_\pi V_\mathcal{P}(\pi)) - V_\mathcal{P}(\pi^*) \leq \varepsilon$.

*Proof.* **Sampling the dataset.** We have sampling access with a fair policy, so every state has positive probability to be visited. Thus, for any finite number $n$, we can almost surely obtain $n$ samples of every state $s$ in finite time. For the Bernoulli process underlying Equation (1), and if the missingness function is simple MAR, this suffices to guarantee that for every state-observation pair, we can obtain the number of samples $n_{s,z}$ required for achieving precision $\varepsilon$ with confidence $\delta_{s,z}$. Similarly, for the Bernoulli process underlying Equation (2), and if the missingness function satisfies positivity, we can also obtain the required number of samples for every state-observation pair. Overall, under the assumptions of the theorem, we can almost surely obtain a dataset in finite time such that it suffices to give PAC guarantees on every state-observation pair.

We remark that this does not even require spending confidence budget as we did in the proofs of Theorems 1 and 2, since there we required to get this dataset within a certain number of histories $n^*$. Here, we only claim that we can get a sufficient dataset in finite time almost surely.

**Obtaining $\widehat{M}$.** The assumptions on the missingness function in the statement of the theorem match those in Theorem 1 or Theorem 2. Hence, given the dataset described in the previous paragraph, we can approximate $\widehat{M}$ in a way such that with probability $\delta$, it is $\varepsilon_M$-precise. Note that here we do not employ the full allowed imprecision $\varepsilon$, but rather a smaller $\varepsilon_M < \varepsilon$, since there will be other sources of error.

**$M$ and $\widehat{M}$ qualitatively agree.** For our technical reasoning, we require that $M(z \mid s) = 0$ if and only if $\widehat{M}(z \mid s) = 0$. We prove both directions separately: If $M(z \mid s) = 0$, then we never observe a sample for $z$ when given $s$, and thus $\widehat{M}(z \mid s) = 0$, as it uses an empirical average (Equations (1) and (2)). If $M(z \mid s) > 0$, as we use a fair sampling process, we almost surely eventually observe $z$ when given $s$, and consequently the empirical average is positive, i.e. $\widehat{M}(z \mid s) > 0$.

It remains to prove that we can *in finite time* conclude that $M$ and $\widehat{M}$ qualitatively agree. This means that we need to be sufficiently certain that if $\widehat{M}(z \mid s) = 0$, this is because indeed $M(z \mid s) = 0$ and not just because we haven't sampled enough yet. For this, we use a proof technique employed in, e.g., Daca et al. (2017): We utilize knowledge of (a lower bound on) the smallest missingness probability $p$. Further, recall that the confidence threshold $\delta$ is distributed over all Bernoulli processes (see Appendices B.2 and B.3). Thus, for each Bernoulli process, we have a confidence threshold $\delta_{s,z}$. Okamoto's inequality (see Appendix B.1) provides an upper bound on the missingness probability that is correct with probability at least $\delta_{s,z}$. Thus, when this upper bound is less than $p$, we can conclude with sufficient confidence that $\widehat{M}(z \mid s) = 0$.

**Utilizing Lemma 2.** Let $\widehat{\mathcal{P}}$ be the approximated missingness-MDP that is exactly $\mathcal{P}$ except for the missingness function, which is $\widehat{M}$ instead of $M$. We have just proven that in finite time we know that with probability $\delta$, $\widehat{M}$ is $\varepsilon_M$-precise and qualitatively agrees with $M$. Thus, it satisfies the assumptions specified in Lemma 2, which is proven below. This key technical lemma shows that the values obtained when following a policy $\pi$ in either the original $\mathcal{P}$ or the approximated $\widehat{\mathcal{P}}$ have a bounded difference.[3] Formally, for every policy $\pi$, we have $|V_{\mathcal{P}}(\pi) - V_{\widehat{\mathcal{P}}}(\pi)| \leq f(\varepsilon_M)$, where $f$ is a monotonically increasing function that depends on $\varepsilon_M$, the precision of $\widehat{M}$.

From this, we obtain two facts: Firstly, since this holds for all policies, it also holds for the supremum over all policies, and thus we can bound the difference in the values of the two missingness-MDPs:

$$|\sup_{\pi} V_{\mathcal{P}}(\pi) - \sup_{\pi} V_{\widehat{\mathcal{P}}}(\pi)| \leq f(\varepsilon_M). \tag{4}$$

Secondly, we can apply the same reasoning to a near-optimal policy in $\widehat{\mathcal{P}}$. For this, let $\varepsilon_{\pi} < \varepsilon$ be a precision smaller than our overall error tolerance, and let $\pi^*$ be an $\varepsilon_{\pi}$-optimal policy in $\widehat{\mathcal{P}}$, i.e.

$$\sup_{\pi}(V_{\widehat{\mathcal{P}}}(\pi)) - V_{\widehat{\mathcal{P}}}(\pi^*) \leq \varepsilon_{\pi}. \tag{5}$$

We remark that $\widehat{\mathcal{P}}$ is a fully specified missingness-MDP, and thus a fully specified POMDP, for which solvers computing $\varepsilon$-optimal policies such as SARSOP (Kurniawati et al., 2008) exist. Using Lemma 2, we obtain the following inequality:

$$|V_{\mathcal{P}}(\pi^*) - V_{\widehat{\mathcal{P}}}(\pi^*)| \leq f(\varepsilon_M). \tag{6}$$

**Implications of the inequalities.** Since we reason about absolute differences, we need to make case distinctions on whether $\sup_{\pi} V_{\mathcal{P}}(\pi) - \sup_{\pi} V_{\widehat{\mathcal{P}}}(\pi) \geq 0$ or not when applying Equation (4). If $\sup_{\pi} V_{\mathcal{P}}(\pi) - \sup_{\pi} V_{\widehat{\mathcal{P}}}(\pi) \geq 0$, then $\sup_{\pi} V_{\mathcal{P}}(\pi) - \sup_{\pi} V_{\widehat{\mathcal{P}}}(\pi) \leq f(\varepsilon_M)$, and by reordering we get $\sup_{\pi} V_{\mathcal{P}}(\pi) \leq \sup_{\pi} V_{\widehat{\mathcal{P}}}(\pi) + f(\varepsilon_M)$. Otherwise, we have $\sup_{\pi} V_{\mathcal{P}}(\pi) < \sup_{\pi} V_{\widehat{\mathcal{P}}}(\pi)$. Together, we can obtain that Equation (4) implies:

$$\sup_{\pi} V_{\mathcal{P}}(\pi) \leq \sup_{\pi} V_{\widehat{\mathcal{P}}}(\pi) + f(\varepsilon_M) \tag{7}$$

Analogously, we can make a case distinction in Equation (6) and obtain that:

$$V_{\widehat{\mathcal{P}}}(\pi^*) \leq V_{\mathcal{P}}(\pi^*) + f(\varepsilon_M) \tag{8}$$

**Combining the inequalities.** To conclude the proof, we use a chain of inequalities.

$$\sup_{\pi} V_{\mathcal{P}}(\pi) \leq \sup_{\pi} V_{\widehat{\mathcal{P}}}(\pi) + f(\varepsilon_M) \qquad \text{(By Equation (7))}$$
$$\leq V_{\widehat{\mathcal{P}}}(\pi^*) + \varepsilon_{\pi} + f(\varepsilon_M) \qquad \text{(By Equation (5))}$$
$$\leq V_{\mathcal{P}}(\pi^*) + f(\varepsilon_M) + \varepsilon_{\pi} + f(\varepsilon_M) \qquad \text{(By Equation (8))}$$

By reordering, we obtain

$$|\sup_{\pi} V_{\mathcal{P}}(\pi) - V_{\mathcal{P}}(\pi^*)| \leq \varepsilon_{\pi} + 2 \cdot f(\varepsilon_M).$$

Hence, since $f$ is a monotonically increasing function, there exists a choice of $\varepsilon_M$ and $\varepsilon_{\pi}$ so that $\varepsilon_{\pi} + 2 \cdot f(\varepsilon_M) \leq \varepsilon$. Intuitively, while the errors incurred by approximating $\widehat{M}$ and by using an approximately optimal policy add up, we can bound the overall maximum error. Thus, we can choose the two precisions so that the overall error criterion is met, and the policy $\pi^*$ is $\varepsilon$-optimal in the original missingness-MDP (with probability $\delta$; with the remaining probability, our sampling was unlucky and $\widehat{M}$ can differ by more than $\varepsilon_M$). This concludes the proof. $\qquad \square$

**Lemma 2** (Bounding the Value-Difference between $\mathcal{P}$ and $\widehat{\mathcal{P}}$)**.** Let $\mathcal{P}$ be a missingness-MDP and $\widehat{\mathcal{P}}$ be a missingness-MDP that differs from $\mathcal{P}$ only in its missingness function, where it uses $\widehat{M}$ instead of $M$. Further, assume that for all states $s \in S$ and observations $z \in Z$, we have $M(z \mid s) = 0$ if and only if $\widehat{M}(z \mid s) = 0$, and moreover $M(z \mid s) = \widehat{M}(z \mid s) \pm \varepsilon_M$. Then, for every policy $\pi$ we have $|V_{\mathcal{P}}(\pi) - V_{\widehat{\mathcal{P}}}(\pi)| \leq f(\varepsilon_M)$, where $f$ is a monotonically increasing function.

---

[3]We highlight that every policy is applicable in both missingness-MDPs, as they only differ in their missingness probabilities, but agree on states, observations, and actions.

*Proof.* **To uncountable MDPs.** Note that both $\mathcal{P}$ and $\widehat{\mathcal{P}}$ are missingness-MDPs, and thus POMDPs. Thus, for each of them, we can construct an uncountable belief MDP with the same value, called $B$ or $\widehat{B}$, respectively. Intuitively, this is achieved by unrolling step-by-step the observation function and all possible beliefs that the agent can have after an action; the transition probabilities in these uncountable MDPs depend on the missingness functions. For a more extensive description, see (Russell & Norvig, 2022, Chapter 16.4.1).

**To finite MDPs.** We consider discounted expected reward, with $\gamma$ the discount factor and $\varrho_{\max} \coloneqq \max_{(s,a) \in S \times A} \varrho(s,a)$ the maximum state reward. As the expected reward is a geometric series, we can bound the reward that can be obtained after $n$ steps from above as follows:

$$\sum_{i=n}^{\infty} \gamma^i \cdot \varrho_{\max} = \gamma^n \cdot \varrho_{\max} \cdot \sum_{i=0}^{\infty} \gamma^i = \frac{\gamma^n \cdot \varrho_{\max}}{1 - \gamma}.$$

For every arbitrarily small precision $\varepsilon_\gamma > 0$, we can thus obtain an $n$ such that the reward after $n$ steps is less than $\varepsilon_\gamma$. Let $B_{\varepsilon_\gamma}$ be the finite MDP obtained from $B$ by only considering states that are reachable within $n$ steps, and analogously define $\widehat{B}_{\varepsilon_\gamma}$. (Note that $n$ is the same for both, since it only depends on $\gamma$ and $\varrho_{\max}$, which is the same for both of them.) The value of these finite belief MDPs differs from the value of the uncountable belief MDPs and thus the original missingness-MDPs by at most $\varepsilon_\gamma$.

**Bounding the difference.** Recall that $B$ or $\widehat{B}$ are the same except for their transition functions, which depend on $M$ and $\widehat{M}$, respectively. Still, by assumption of the theorem $M$ and $\widehat{M}$ qualitatively agree, i.e. $M(z \mid s) = 0$ if and only if $\widehat{M}(z \mid s) = 0$. Hence, the graph structure of $B$ or $\widehat{B}$ is the same. Thus, the only difference are small perturbations of individual transition probabilities by at most $\varepsilon_M$.

It remains to show the following: Given two finite MDPs that are the same except for small perturbations of the transition probabilities, but where the supports of the the transition functions are the same, provide a bound on the difference in their value. Such a result exists in the literature, namely in Meggendorfer et al. (2025), or more precisely in the extended version of that paper (Meggendorfer et al., 2024, Lemma 5). It remains to show that our setting indeed satisfies the assumptions of (Meggendorfer et al., 2024, Lemma 5).

- "For every closed constant-support RMDP": Their claim applies to robust MDPs that are closed constant-support. A robust MDP is an MDP whose transitions are not probability distributions, but rather sets of possible values, see (Meggendorfer et al., 2025, Section 2). In our case, instead of considering the concrete MDPs $B_{\varepsilon_\gamma}$ and $\widehat{B}_{\varepsilon_\gamma}$, we consider the robust MDP that arises when considering an $\varepsilon_M$-interval around every missingness probability $M(z \mid s)$. This robust MDP contains both $B_{\varepsilon_\gamma}$ and $\widehat{B}_{\varepsilon_\gamma}$ as instantiations.

- "For every pair of agent and environment policy": An agent policy in this setting is exactly the agent policy in ours, so (Meggendorfer et al., 2024, Lemma 5) applies to all policies. An environment policy is the policy that chooses the instantiation of the transition function, i.e. the exact missingness probabilities from the set of all that differ by at most $\varepsilon_M$ in our setting.

- "Total-reward objectives:" (Meggendorfer et al., 2024, Lemma 5) concerns *undiscounted* total-reward or mean payoff objectives. Undiscounted total-reward generalizes discounted expected reward, using the standard construction which adds an edge transitioning with probability $\gamma$ to a dedicated sink state to every transition. Thus, the lemma is applicable to the objective in our setting.

- "The value function is continuous w.r.t. the environment policy": This is the claim of (Meggendorfer et al., 2024, Lemma 5). More formally, if the environment chooses missingness probabilities differently with some deviation $\varepsilon_M$, then the deviation in the value between the two instantiations is bounded by some monotonically increasing function $g(\varepsilon_M)$. This is exactly the claim we require, since it means that for all agent policies $\pi$ and all missingness functions $\widehat{M}$ that are $\varepsilon_M$-close to $M$, we have $|V_{B_{\varepsilon_\gamma}}(\pi) - V_{\widehat{B}_{\varepsilon_\gamma}}(\pi)| \leq g(\varepsilon_M)$.

  We also argue that $g$ can be effectively computed, as it depends on the size of the state space, the reward function, and the minimum occurring transition probability, all of which are known to us (recall that Theorem 3 assumes knowledge of a lower bound on the minimum missingness

probabilities). The concrete way of deriving the distance is provided on (Meggendorfer et al., 2024, page 17).

**Putting it all together.** Our goal is to show that we can compute an $f$ such that for all policies $\pi$ we have: $|V_{\mathcal{P}}(\pi) - V_{\widehat{\mathcal{P}}}(\pi)| \leq f(\varepsilon_M)$. The following chain of equations proves our goal:

$$|V_{\mathcal{P}}(\pi) - V_{\widehat{\mathcal{P}}}(\pi)| = |V_B(\pi) - V_{\widehat{B}}(\pi)|$$

(Using the uncountable belief MDPs)

$$\leq |B_{\varepsilon_\gamma}(\pi) - V_{\widehat{B}_{\varepsilon_\gamma}}(\pi)| + \varepsilon_\gamma$$

(Using the finite MDPs; decreasing both values by
at most $\varepsilon_\gamma$ increases the difference by at most $\varepsilon_\gamma$)

$$\leq g(\varepsilon_M) + \varepsilon_\gamma$$

(By bounding the difference).

For simplicity of presentation, we choose $\varepsilon_\gamma = \varepsilon_M$, and thus setting $f(\varepsilon_M) \coloneqq g(\varepsilon_M) + \varepsilon_M$ concludes the proof. □

## C BENCHMARKS

Here we describe our benchmarks. We provide a detailed description of the benchmarks as well as the parameters for running the experiments.

### C.1 DESCRIPTION

***ICU.*** This benchmark, inspired by prior clinical decision-making models (Johnson et al., 2022; Pollard et al., 2018; Thoral et al., 2021; Hyland et al., 2020), simulates a doctor treating a patient with an infection that progresses stochastically over time. The state of the patient consists of the *infection severity*, the *temperature*, and the *heart rate*. The infection causally influences both the heart rate and the temperature.

The doctor has an option to wait, to administer costly antibiotics that reduce the infection severity, or to order a test, which is a measuring action that may reveal the infection severity. The reward function penalizes high infection levels as well as costly interventions (ordering a test and administering antibiotics). Thus, the doctor's objective is to maintain the patient's infection severity at low levels by administering antibiotics only when necessary. For ease of modeling, the state space also includes the value of the last test ordered.

We evaluate three different missingness functions $M$, corresponding to distinct missingness functions, illustrated in the m-graph in Figure 4. In all cases, the heart rate and the infection severity may be missing, whereas temperature and the last test ordered are always observed. The success rate of the

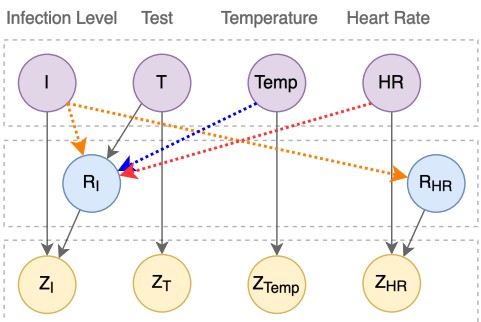

Figure 4: The m-graphs for the ICU benchmark describing missingness functions of types *simple MAR* (gray + blue), *identifiable MNAR* (gray + red) and *unidentifiable MNAR* (gray + red + orange). Causal dependencies between the state features were omitted for clarity.

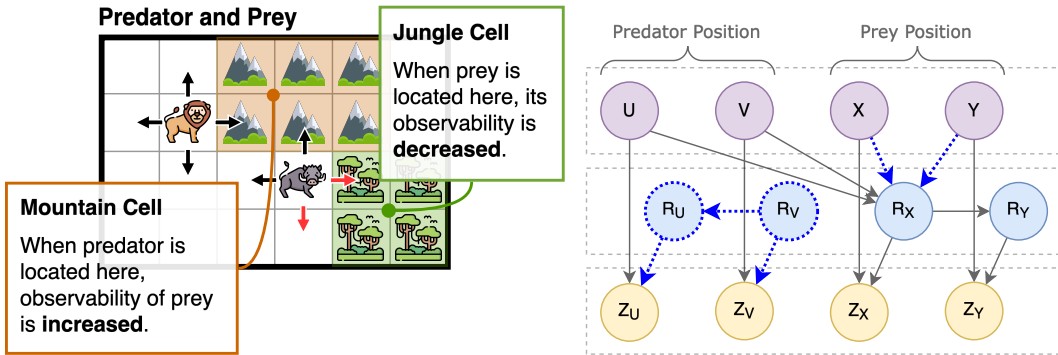

Figure 5: **left:** The *Predator* benchmark, where the predator (lion) is the agent trying to catch its prey (boar). Predator and prey can move in all four cardinal directions, where prey chooses an action that increases the distance to the predator (red arrows). **right:** The m-graphs for the predator and prey benchmark describing missingness functions of types *simple MAR* (gray), *identifiable MNAR* (gray + blue). Causal dependencies between the state features were omitted for clarity.

test that reveals the infection severity may depend on different features, resulting in the following missingness functions. **(1) Simple MAR**, where the success rate only depends on the (always observed) temperature. **(2) MNAR (id.)**, where the success rate only depends on the (not always observed) heart rate, resulting in an identifiable MNAR function without self-censoring and satisfying the positivity assumption. **(3) MNAR (unid.)** is an extension of **MNAR (id.)**, where the infection severity influences the test success rate, introducing self-censoring and thus making the function unidentifiable.

***Predator.*** This benchmark is a variant of the *Tag* benchmark from Pineau et al. (2003), where an agent (in our case, a predator) is tasked with chasing a partially hidden target (a prey) in a 2D grid environment. The prey senses the predator and usually moves away from it; in case multiple directions lead away from the predator, the prey chooses uniformly at random. The predator's movement is deterministic (dictated by the policy), but moving in an intended direction may randomly fail due to terrain conditions. Predator obtains a flat reward upon catching the prey, and thus the discounting incentivizes catching the prey as soon as possible.

The environment may feature three distinct *biomes* – plains, mountains, or jungles – that influence the predator's observability of the prey, see Figure 5, and thus define the missingness function. We investigate the following three variants thereof. **(1) MCAR**, which features only one type of terrain, i.e., the prey is observed with constant probability. **(2) simple MAR**, where the environment features plains as well as mountains from which the predator has a higher chance of observing its target. **(3) MNAR (unid.)**, where the prey has an option to hide in jungle cells, introducing self-censoring of its position. We stress that when the predator loses track of the prey, both features corresponding to $x$ & $y$ coordinates of the prey go missing simultaneously, modeled by dependencies between missingness indicators $R_x$ & $R_y$. The dependence between the missingness indicators is a key difference from the ICU benchmark.

## C.2  EXPERIMENTAL SETUP

**Technical Setup.** For all experiments, we used high-performance workstations equipped with an AMD Ryzen ThreadRipper PRO 5965WX (24-core, 3.8GHz) CPU, 512 GB ECC DDR4 RAM, and a 2 TB PCIe 4.0 NVMe SSD.

**Simulating trajectories.** For both benchmarks, we used a discount factor of $\gamma = 0.95$. We considered dataset sizes $|\mathcal{D}| \in \{10, 50, 100, 500, 10^3, 5 \cdot 10^3, 10^4, 10^5, 10^6, 10^7\}$. To obtain a dataset containing $|\mathcal{D}|$ samples, we simulated finite trajectories until their lengths summed up to $|\mathcal{D}|$. A trajectory is terminated when it reaches a terminal state (only for the *Predator* benchmark, when the predator catches the prey) or if its length exceeds $L = \left\lceil \log_\gamma \frac{(1-\gamma) \cdot 10^{-3}}{\varrho_{\max}} \right\rceil$, where $\varrho_{\max} := \max_{s,a} \varrho(s, a)$. Here, $L$ denotes the smallest integer satisfying $\sum_{k=L}^{\infty} \gamma^k \cdot \varrho_{\max} < 10^{-3}$, i.e. a time step after which

the maximum discounted cumulative reward cannot exceed $10^{-3}$. For each dataset size $|\mathcal{D}|$, we generated 20 independent datasets of this size.

**Timeouts & precision.** For the baselines, we used the timeout of 5 minutes when solving the POMDP (to obtain $\pi^*$ and $\pi^{M_u}$) and the same timeout to evaluate the resulting policy (or $\pi^{\mathrm{md}}$). To obtain a policy $\hat{\pi}$ by solving the corresponding $\widehat{\mathcal{P}}$, we used a timeout of 3 minutes and evaluated $\hat{\pi}$ for 2 minutes. In all cases, solving was additionally allowed to terminate upon reaching the relative precision of $10^{-3}$.

