# OpenReview forum: "Missingness-MDPs: Bridging the Theory of Missing Data and POMDPs"
_ICLR.cc/2026/Conference — ICLR 2026 Conference Withdrawn Submission_

### Official Review · Reviewer_BvRg · 2025-10-28

**Soundness:** 2
**Presentation:** 2
**Contribution:** 2
**Rating:** 4
**Confidence:** 3

**Summary:**

- This work addresses a particular type of POMDP, called a "miss-MDP," where certain state features are missing at random or through dependencies with other state features, and proposes an approach for estimating the missingness function from a historical dataset and computing a near-optimal PAC policy.
- They evaluate the proposed algorithm (and its variants) using an ICU dataset and a toy domain under different assumptions about the missingness function and compare their performance to the oracle and the uniform action-selection baselines.

**Strengths:**

- Originality and significance: This work extends existing research on POMDPs and MDPs with missing state observations by introducing an estimation method for the missingness function under different assumptions about missingness.

- Quality: The paper includes both a toy domain and a clinical dataset.

- Clarity: Overall, the writing is clear, with a few outstanding questions regarding the distinction between MNAR and non-simple MAR (which the reviewer elaborates on in the Weaknesses section). Figure 2 effectively illustrates different cases of missingness and is helpful to include.

**Weaknesses:**

Originality and significance:
- While the problem itself is interesting and important, the analysis and applicability of the proposed algorithm are limited to tabular states, and it is unclear how the method could be extended to high-dimensional states. Adding to this point, the experimental domains have fairly small state spaces (2 for the predator task and 4 for the ICU task) whereas in real-world settings, where the motivation for POMDPs and missingness comes from, state observations are typically high-dimensional, which limits the applicability and significance of this work.

- Although the inclusion of a PAC analysis is valuable, it is fairly standard and not technically novel enough to constitute an independent contribution.

Experiments:
- The current experiments include only variants of the authors’ own algorithms (besides the uniform and the oracle baselines). The authors could consider modifying or including baselines from prior work, such as deep variational methods by Igl et al., 2018, since it is difficult to calibrate the empirical advantages without comparisons to other approaches, even if those methods are based on different assumptions about missingness. (Igl et al., 2018. Deep variational reinforcement learning for POMDPs. There may be other works that can be adapted to the problem setup of this paper and can be included as comparisons.)

Clarity:
- The distinction between non-simple MAR and MNAR discussed in Section 4.2 (where the authors state that the non-simple MAR case satisfies the three conditions of independence, no self-censoring, and positivity) is unclear. In particular, the example provided for "MAR but not simple MAR" is: “Now, the missingness probability of feature 1 depends on the value of feature 2 (only if observed), while feature 2 itself misses with probability 0.5.” This example appears to correspond to the MNAR case shown in Figure 2, where $S_2$ affects $R_1$ (so feature 1 depends on the value of $S_2$ if it's observed), while $S_2$ itself can be missing as suggested by $R_2$.

- This confusion continues with Lines 200-201, where the authors describe MNAR as cases where "missingness probabilities may depend on the values of missing functions" and provide a self-censoring example for MNAR. However, in the later sections (particularly 4.2), MNAR is described as non-self-censoring, independent, and positive. The authors could consider providing a different example of MNAR that satisfies these conditions to clearly distinguish between non-simple MAR and MNAR. Without further clarification, the description of MNAR may mislead readers into interpreting it as cases where missingness depends on the feature’s value (e.g., when temperature is too high or too low to be recorded properly, making the missing value itself informative about the measurement).

**Questions:**

- Question about the distinction of MNAR in section 4.2 and non-simple MAR is raised in Weaknesses.

---

> ### Author Response · Authors · 2025-11-26
> **Rebuttal by Authors**
>
> We thank the reviewer for their comments on the submission. Below, we address some of the mentioned weaknesses and provide answers to their questions.
>
> > **“Although the inclusion of a PAC analysis is valuable, it is fairly standard and not technically novel enough to constitute an independent contribution.”**
>
> The key contribution of our paper is to combine the fields of planning under uncertainty and missingness theory, in particular showing that, for certain types of missingness functions, we can provide efficient algorithms. The PAC analysis is the theoretical foundation of the latter point. Since we (1) are, to the best of our knowledge, the first to consider this particular combination of fields and (2) show the empirical and theoretical advantages of exploiting missingness patterns as opposed to mere POMDP reasoning, we believe that the approach and concept do constitute a valuable contribution to the field of POMDP planning.
>
> > **Clarification regarding the distinction between “non-simple MAR” and MNAR.**
>
> The MNAR graph in Figure 2 is indeed MNAR. Whether observed or not, $S\_2$ may influence $R\_1$, violating the MAR requirement (where feature values may influence the missingness probabilities only when observed).
> If the graph were to represent “non-simple” MAR, and $S\_2$ were to influence $R\_1$ only when observed, we would require an additional edge between $R\_2$ and $R\_1$.
> To improve clarity, we added an arrow from $S\_2$ to $R\_2$ to the MNAR graph, introducing self-censoring, which **always** results in an MNAR missingness function.
> We also added clarifications to Example 3 and to the beginning of Section 4.2, stating that AIMI’s assumptions correspond to a subset of MNAR missingness functions.
>
> > **The experiments do not compare the proposed algorithms to other baselines, such as deep variational reinforcement learning.**
>
> We did not consider RL baselines, in particular deep variational RL, since our methods make use of a *known* transition function and, most importantly, leverage the assumption about the structure of the missingness (observation) function. Further, our methods come with guarantees regarding the optimality of the approximation of the missingness function and the resulting policy. Thus, our methods are incomparable to deep variational RL, as they rely on stronger assumptions, but also provide stronger guarantees.

---

### Official Review · Reviewer_bJ4V · 2025-10-30

**Soundness:** 2
**Presentation:** 3
**Contribution:** 2
**Rating:** 2
**Confidence:** 3

**Summary:**

This paper formalizes miss-MDPs and gives PAC-type guarantees for learning missingness under simple MAR and MNAR with independent incidators and no self-censoring, then use a standard POMDP method. Experiments suggest convergence to optimal policy with reasonable data sizes.

**Strengths:**

- Formalize the miss-MDPs to bridge between missing data taxonomy and sequential decision making.
- Shows under MAR, belief updates do not depend on missingness probabilities.

**Weaknesses:**

- AsMAR estimates the set $\hat{I}_{always}$, but Theorem 1's PAC statement does not condition on this set.
- Counting set for AIMI is too restrictive. $Z_s^{i,r_i}$ requires all features $j\neq i$ be observed and exactly match $s$, discarding any sample where any other component is missing. This can devastate sample efficiency, even under positivity, and the paper gives no guidance on the resulting rates.
- By design, $M$ depends only on $s$ (not $a$ or $t$), and your POMDP observation in prelims is also action-independent. Many practical missingness processes (e.g., clinical testing) are action-selected.

**Questions:**

See weakness

---

> ### Author Response · Authors · 2025-11-26
> **Rebuttal by Authors**
>
> We thank the reviewer for their comments on the submission. Below, we address some of the mentioned weaknesses and provide answers to their questions.
>
> > **Theorem 1 does not condition on I\_hat\_always.**
>
> **We would like to clarify this misunderstanding.** `I_hat_always` is the estimate of `I_always` derived from the dataset, whereas `I_always` is part of the missingness function and its definition of simple MAR. Therefore, a correctness statement should condition on `I_always` and conclude on `I_hat_always`. That is precisely the case: Theorem 1 conditions on the missingness function being simple MAR (and thus also on `I_always`), while concluding that the error in `M_hat` is bounded, which implicitly includes `I_hat_always`.
>
> > **AIMI’s restrictive counting set takes away from sample efficiency.**
>
> While it might seem that AIMI is sample inefficient as it requires all features but one to be observed;
> (1) this is necessary for the correctness of counting methods for AIMI’s type of missingness functions.
> And (2) upon closer inspection, we can see that we approximate observation probabilities for each feature independently, resulting in a factorized distribution. Approximating a factored distribution requires fewer samples than approximating a joint distribution, which offsets the inefficiency of discarding samples. Consequently, we observe a reasonable convergence rate for AIMI in the ICU MNAR (id.) experiment in Figure 3\.
> Also note that for an observation originating from a state, all its non-missing feature values are equal to the state’s features.
>
> > **The missingness function does not take actions or time into account.**
>
> **Answer:** We have added a clarification in Section 3 as missingness functions are (1) time-invariant and (2) may take actions into account.
> (1) Missingness functions act as a structured observation function, which in the underlying POMDP model is typically assumed to be Markovian and therefore time-invariant.
> (2) Furthermore, action-dependent observation functions (and, consequently, missingness functions) can be w.l.o.g. represented as state-dependent functions by augmenting the state space of the POMDP via standard transformation (Chatterjee et al., 2016).
>
> **References**
>
> * Krishnendu Chatterjee et al. Optimal cost almost-sure reachability in POMDPs. In Artificial Intelligence, 2016\.

---

### Official Review · Reviewer_mvJv · 2025-11-01

**Soundness:** 3
**Presentation:** 3
**Contribution:** 2
**Rating:** 4
**Confidence:** 3

**Summary:**

This paper presents a subclass of POMDPs where the state can be observed with certain components missing. Standard assumptions on missingness from the theory of missing data are investigated. Theoretical analysis is provided for estimating the missingness model given a dataset of trajectories sampled by running a so-called fair policy, and the consistency of a policy computed using the estimated model. The theoretical insights are empirically validated on two simple problems.

**Strengths:**

* Incomplete state observations can arise in practice. Formalizing and analyzing this are interesting.
* The writing is clear and easy to follow.

**Weaknesses:**

* The main contribution of the paper is in the theoretical analysis, but this applies to POMDPs with finite state and action spaces only, which is not particularly surprising.
* Empirical experiments are only done on simple small problems.
* A minor comment is that PAC as used in this paper is different from the standard concept of PAC as introduced by Valiant, which can be confusing.
* Another minor comment is that "observation" and "trajectory" are used synonymously at some places (e.g. the abstract mentions "a dataset of observations", while it means a dataset of trajectories).

**Questions:**

* Does the analysis cover the case when some states are not reachable from some other states?
* In Theorem 1 and Theorem 2, presumably the result holds for not just a particular $n^{\*}$, but for any sufficiently large $n^{\*}$, is it?
* The experiments show that exact missingness function can be estimated. It is surprising that probabilities can be estimated without any error using random datasets. Did I miss something?

---

> ### Author Response · Authors · 2025-11-26
> **Rebuttal by Authors**
>
> We thank the reviewer for their comments on the submission. Below, we address some of the mentioned weaknesses and provide answers to their questions.
>
> > The submission’s definition of PAC differs from the definition of Valiant.
>
> We added an explanation to our submission, after Theorem 3 (Section 4.3) on computing $\varepsilon$-optimal policies: *“Note that we use the notion of PAC guarantee that is common in statistical model checking (Brazdil et al., 2025; Ashok et al., 2019). This is inspired by, but slightly different from the original definition of Valiant (1984), as we return in finite time a policy that performs close to optimal with high probability.”*
>
> > Does the analysis cover the case when some states are not reachable from some other states?
>
> Yes, we do not assume full connectivity between states and allow for sparse transition functions.
>
> > “In Theorem 1 and Theorem 2, presumably the result holds for not just a particular n*, but for any sufficiently large n*, is it?”
>
> Yes, any number larger than n* will satisfy the PAC conditions.
>
> > In the experiments, it is surprising that probabilities can be estimated without any error using random datasets.
>
> In our experiments, in Figure 3, as the size of the dataset increases, the error of the approximated missingness function (under the right assumptions and with an appropriate algorithm) approaches zero arbitrarily closely, without being exactly zero. Note that the dataset is generated by a fair policy that eventually explores all reachable states.

---

### Official Review · Reviewer_xCFs · 2025-11-03

**Soundness:** 3
**Presentation:** 2
**Contribution:** 2
**Rating:** 2
**Confidence:** 4

**Summary:**

This paper presents a novel formalism, so-called missingness Markov decision processes (miss-MDPs), which is a subclass of partially observable MDPs (POMDPs). The main contribution of the paper is to introduce three types of missingness functions, which are missing completely at random (MCAR), missing at random (MAR),
and missing not at random (MNAR), into MDPs. The authors prove finite-time PAC guarantees for estimating missingness functions and ensuring \epsilon-optimal policies.

**Strengths:**

The paper gives an interesting formalism for a subclass of POMDPs and has PAC-bounds on learning policies for miss-MDPs.

**Weaknesses:**

Overall, I believe the paper has an interesting theoretical approach, but it lacks motivation in using the miss-MDP formalism instead of POMDPs, as the authors did not show a specific computational advantage either in theory or in practice. I will have some questions in my review.

**Questions:**

What is the main advantage of using miss-MDPs instead of simply treating them as a special observation function in POMDPs?

What new capabilities or insights does the miss-MDP formalism provide that POMDPs do not? You only showed finite-time PAC bounds in your paper.

Can you provide some examples where this approach may provide better theoretical and practical guarantees compared to just using POMDPs?

How does admittability affect learning or belief updates later? Could multiple states admit the same observation z? If so, how does this ambiguity influence identifiability?

What is the rate at which the error in \hat{M} translates into policy suboptimality \epsilon? Is it possible to establish bounds as a function of the number of states, observations, and the missingness function?

---

> ### Author Response · Authors · 2025-11-26
> **Rebuttal by Authors**
>
> We thank the reviewer for their comments on the submission and would like to address some of the mentioned weaknesses as well as give answers to their questions.
>
> > **What is the advantage/capabilities/insights of miss-MDPs over POMDPs?**
>
> The key advantage of miss-MDPs is that their observation functions, referred to as missingness functions, exhibit a specific structure. In particular, learning the missingness function is possible for several classes of missingness functions (as detailed in the submission), while learning observation functions in general is intractable (Liu et al., 2022a; Liu et al., 2022b).
>
> > **“Could multiple states admit the same observation z? If so, how does this ambiguity influence identifiability?”**
>
> Yes, multiple states can admit the same observations; e.g. $s=(0,0)$ and $s’=(1,0)$ both admit the observation $z=(\\bot, 0)$. (Un-)identifiability issues in miss-MDPs arise precisely due to this ambiguity. Identifiability is extensively documented in the missing data literature (Little and Rubin, 2019; Tsiatis, 2006; Bhattacharya et al., 2020), and our work therefore does not focus on studying particular identifiable instances. To further illustrate, a missingness function where observations have unique admitting states would be trivially identifiable.
>
> >**”What is the rate at which the error in $\\hat{M}$ translates into policy suboptimality $\epsilon$? Is it possible to establish bounds”?**
>
> As discussed in Appendix B.4, the correspondence between the error in $\\hat{M}$ and the suboptimality of the policy is given by a function $f$ (line 895). Explicit theoretical bounds could be constructed via Okamoto’s inequality or similar principles. We can provide these in the camera-ready submission. Nevertheless, we empirically analyze this correspondence, as shown in Figure 3, where a lower error in $\\hat{M}$ leads to a better policy when using the appropriate algorithm.
>
> **References**
>
> * Qinghua Liu et al. When is partially observable reinforcement learning not scary? In Proceedings of Machine Learning Research, 2022\.
> * Zeyu Liu et al. A Machine Learning–Enabled Partially Observable Markov Decision Process Framework for Early Sepsis Prediction. In INFORMS Journal on Computing, 2022\.
> * Roderick Little and Donald Rubin. Statistical Analysis with Missing Data, Third Edition. Wiley Series in Probability and Statistics. Wiley, 2019\.
> * Anastasios A Tsiatis. Semiparametric Theory and Missing Data. Springer Series in Statistics. Springer, 2006\.
> * Rohit Bhattacharya et al. Identification In Missing Data Models Represented By Directed Acyclic Graphs. In UAI 2020\.

---

### Author Response · Authors · 2025-11-26
**Rebuttal by Authors**

We thank the reviewers for their detailed assessment of the paper.
All reviewers consider the concept of missingness in observations interesting and a strength of our paper. Further, reviewers `mvjv` and `BvRg` note that the paper is written in a clear and easy-to-follow manner.

Below, we address major critical points raised by the reviewers, followed by a summary of the individual answers.

1. **Reviewers `mvjv` & `BvRg` note limitations of finite state and action spaces, raising doubts about scalability and real-world applicability.**
  The understanding that sensors may dynamically obscure state features, referred to as missingness, has (to the best of our knowledge) not been explicitly studied for POMDPs.  In this submission, we lay the theoretical groundwork required to understand the intricacies and benefits of combining the theory of missingness with POMDP planning. We deliberately consider discrete/finite state and action spaces, as we are interested in providing optimality guarantees. Further, the chosen scales of our problems are sufficient to answer our research questions, such as how the type of missingness function influences the learnability (see Figure 3).

2. **Reviewer `xCFs` inquires about the advantage of miss-MDPs over POMDPs.**
  In miss-MDPs, the missingness function is a structured observation function. In POMDPs, learning the observation functions is, in general, intractable (Liu et al., 2022a; Liu et al., 2022b). Using the miss-MDP formalism, we identify missingness functions that are learnable and for which we provide learning algorithms.

The following points summarize several of our answers, which will be provided in detail in the individual responses.

* Reviewer `xCFs` inquired how the missingness function’s approximation error translates into policy suboptimality. We answer that theoretical bounds would directly follow from Theorems 1 and 3, which we can provide in the camera-ready submission. We refer to our experiments, which empirically demonstrate that a lower error in the missingness function leads to a better policy when using the appropriate algorithm
* Reviewer `mvjv` requested additional clarification regarding our usage of the term probably approximately correct (PAC) and its relation to Valiant’s original definition (Valiant, 1984). We have added a concise explanation of this distinction in Section 4.3.
* Reviewer `bJ4V`  suggested that Theorem 1 may contain an error; this is not the case. We provided a detailed clarification in the response.
* Reviewer `bJ4V` also assumed that the missingness function in miss-MDPs is both action-independent and time-invariant. We clarified in our response and in the revised submission that we allow **action-dependent** missingness functions, and that time-invariance follows directly from the Markovian structure.
* Reviewer `bJ4V` further noted that one of our algorithms, AIMI, appears sample-inefficient. While we acknowledge their concern, we highlight a mitigating aspect and refer to Figure 3, which empirically shows that AIMI converges within a reasonable number of samples.
* Reviewer `BvRg` commented that the differences between “non-simple” MAR and MNAR missingness functions are difficult to follow. We have improved and expanded the corresponding explanations.
* Reviewer `BvRg` suggested adding a baseline based on deep variational RL. We explain that such a baseline is not directly comparable, as our methods assume access to the transition function and leverage structural assumptions of the missingness function.

We thank the reviewers for their feedback, which helped improve the clarity of our submission for the readers. We uploaded a new submission where changes are highlighted in bold and green text.

**References**

* Qinghua Liu et al. When is partially observable reinforcement learning not scary? In Proceedings of Machine Learning Research, 2022\.
* Zeyu Liu et al. A Machine Learning–Enabled Partially Observable Markov Decision Process Framework. In INFORMS Journal on Computing, 2022\.
* Leslie G. Valiant. A theory of the learnable. In Communications of the ACM, 1984\.

---

### Note · Authors · 2026-01-08

**Comment:**

We appreciate the reviewers' time and feedback, which will be valuable for improving our work, but we have decided to withdraw this paper from consideration at the conference as interaction with the reviewers is/was not possible anymore after the OpenReview incident.
Thank you again for your comments.

**Withdrawal Confirmation:**

I have read and agree with the venue's withdrawal policy on behalf of myself and my co-authors.